# Flavonoid-Rich Extract from *Bombyx batryticatus* Alleviate LPS-Induced Acute Lung Injury via the PI3K/MAPK/NF-κB Pathway

**DOI:** 10.3390/ijms262412057

**Published:** 2025-12-15

**Authors:** Wenkai Li, Sifan Song, Wenyong Li, Jing Chen

**Affiliations:** College of Basic Medical Science, Zunyi Medical University, Zunyi 563002, China; liwenkai2024@163.com (W.L.); sifansong773@163.com (S.S.); liwenyong@163.com (W.L.)

**Keywords:** *Bombyx batryticatus*, flavonoids, acute lung injury, anti-inflammatory mechanism, oxidative stress, PI3K/MAPK/NF-κB pathway

## Abstract

Treating acute lung injury (ALI) presents significant challenges due to adverse drug reactions. This study systematically explored the protective effects and mechanisms of a flavonoid-rich extract from *Bombyx batryticatus* (FBB), a traditional Chinese medicine, in combating ALI. Through UPLC-MS/MS analysis, we identified 163 flavonoid components in FBB for the first time, including flavonoids, flavonols, and chalcones. Unlike single-component flavonoid therapies, FBB provides synergistic regulation across multiple targets and pathways. Network pharmacology predictions, supported by experimental validation, revealed that FBB primarily suppresses the expression of inflammatory factors (IL-1β, IL-6, TNF-α) and oxidative stress markers (iNOS, COX-2) by modulating the PI3K/Akt, MAPK, and NF-κB signaling pathways. FBB inhibits pro-inflammatory responses and upregulates chemokine receptors like Ccr1 and Ccr2, along with IL-2Rb, at the transcriptional level. This suggests its potential to promote inflammation resolution and tissue repair through immune microenvironment remodeling, rather than mere immunosuppression. Additionally, FBB demonstrated significant anti-apoptotic effects both in vitro and in vivo, effectively reducing pulmonary edema and vascular permeability. Its complex composition and multi-pathway synergistic mechanisms offer broader regulatory potential and unique therapeutic advantages in treating ALI compared to single flavonoid compounds or conventional hormone drugs like dexamethasone (DEX). This study reveals a novel mechanism by which FBB, a multi-component natural drug, exerts therapeutic effects in ALI, providing a theoretical and experimental foundation for developing flavonoid-based compound preparations from traditional Chinese medicine.

## 1. Introduction

Acute lung injury (ALI) is a severe respiratory disease caused by a variety of etiologies. Its pathological features include impaired alveolar-capillary membrane integrity, leading to the infiltration of protein-rich fluid into the interstitium and alveolar spaces, which in turn causes a decrease in arterial oxygen saturation [1,2,3]. As a pre-stage of acute respiratory distress syndrome (ARDS), ALI has become an important cause of high morbidity and mortality in intensive care units. In pathophysiology, ALI/ARDS goes through several pathophysiological stages [4,5,6]. The first is the exudative stage, in which a cytokine storm in the membrane exacerbates inflammation, promotes neutrophil recruitment and harmful mediators. Local alveolar macrophages secrete TNF-α and IL-1, stimulating further production of IL-6 and IL-8, thereby attracting inflammatory cells and altering alveolar-capillary barrier permeability, thus aggravating pulmonary edema [5,7,8]. Disruption of the coagulation pathway leads to the formation of pulmonary microthrombi, which exacerbates the inflammatory response and causes systemic multi-organ failure [4,5]. This is followed by a fibrotic phase, characterized by reduced alveolar edema, the presence of macrophages and neutrophils, and the regeneration of type II cells. However, some patients may progress to the fibrotic stage, with high mortality and prolonged mechanical ventilation [5].

Sepsis is a major factor in the development of ARDS, and its mechanisms are not limited to direct microbial toxicity but involve multiple pathways, including immune dysregulation, disruption of the endothelial and epithelial barriers, programmed cell death, oxidative stress, and epigenetic regulation [9,10]. After infection initiates the immune response, signaling pathways such as TLR4/NF-κB are activated, releasing pro-inflammatory cytokines such as TNF-α, IL-6, and IL-1β, causing pulmonary microvascular dysfunction and inflammatory cell infiltration, and disrupting the endothelial barrier [11,12,13,14,15]. Advanced glycation end products (RAGE) signaling receptors and the phosphoinositol 3-kinase/protein kinase-B (PI3K/AKT) pathway impair oxidative stress and inflammation, thereby damaging endothelial cells and alveolar epithelial cells. In addition, activation of the NOD-, LRR- and pyrin domain-containing protein 3 (NLRP3) inflammasome and its subsequent pyroptosis contribute to the release of inflammatory mediators, thereby exacerbating lung injury [16,17]. Loss of function of vascular endothelial cadherins and dysregulation of tight junction proteins such as Claudin-4 and Claudin-5 further increase alveolar-capillary permeability, leading to pulmonary edema [8,18,19,20]. In addition, Oxidative stress mainly generates reactive oxygen species through NADPH oxidase subtypes such as NOX1/NOX2, which damage the pulmonary vascular barrier and mitochondrial function [21,22,23].

Currently, drug treatment for ALI primarily involves glucocorticoids, such as dexamethasone (DEX), prednisolone, and ulinastatin [24]. However, these drugs may cause adverse reactions such as coagulation disorders, gastric ulcers, and osteoporosis. While strategies such as lung-protective ventilation can improve patient prognosis, current treatment remains primarily supportive [25,26,27]. Because the pathogenesis of ALI/ARDS is complex, involving the abnormal activation of various immune cells such as dendritic cells, NK cells, macrophages, and neutrophils, gut microbiota dysbiosis, and the cross-regulation of multiple inflammatory signaling pathways such as mTOR, NF-κB, TLR, and JAK-STAT, and is often accompanied by cytokine storms, single-target therapy often has limited efficacy [28]. Therefore, a multi-target strategy integrating immune regulation (such as intervening in cell polarization and inhibiting NET formation), signaling pathway inhibition (such as targeting TLR/NF-κB and MAPK), and gut–lung axis regulation (such as gut microbiota transplantation) holds promise for more comprehensive control of the inflammatory response, reduction of lung damage, and improvement of prognosis through synergistic effects, representing an important direction for future treatment. In the future, it is urgent to further explore intervention methods targeting key pathways related to ALI and develop safer treatment strategies to improve survival rates and clinical outcomes.

Traditional Chinese Medicine (TCM) has a unique theoretical framework and a wealth of practical experience in treating and preventing respiratory disorders. Recent clinical and preclinical studies have shown that Chinese herbal formulations and their bioactive components offer multi-target and multi-pathway regulatory advantages in treating ALI [29,30,31]. Notably, the distinctive capacity of TCM to modulate immune balance and attenuate inflammatory responses has promising therapeutic potential for sepsis-associated ALI. *Bombyx batryticatus*, a TCM material first documented in Shennong’s Classic of Materia Medica, reportedly contains abundant bioactive compounds, including flavonoids, proteins, and polysaccharides [32]. Modern pharmacological studies have shown that these compounds have therapeutic effects on inflammation and oxidative stress as well as immunomodulatory functions [33,34,35]. Most notably, clinical and experimental research data have shown that *B. batryticatus* exhibits therapeutic potential for respiratory conditions, including asthma and chronic obstructive pulmonary disease (COPD) [36,37].

Flavonoids possess a distinct chemical structure and demonstrate significant anti-inflammatory and antioxidant properties [38,39]. These properties enable them to effectively scavenge endogenous free radicals and reactive oxygen species, thereby mitigating cell damage and reducing inflammatory responses induced by oxidative stress [40,41]. Flavonoids achieve their anti-inflammatory effects by selectively regulating key transcription factors, which subsequently alter inflammatory gene profiles. This molecular mechanism leads to the downregulation of pro-inflammatory cytokines such as IL-1β, IL-6, and TNF-α, while promoting the expression of anti-inflammatory factors like IL-10 [41,42,43,44]. Due to these diverse pharmacological effects, flavonoids emerge as promising natural therapeutic candidates for various inflammatory conditions, including arthritis, inflammatory bowel diseases, and pulmonary inflammation [36,45]. Meanwhile, numerous studies have shown that traditional Chinese medicine flavonoid extracts have protective or therapeutic effects against ALI. For example, total flavonoids from the aerial parts of *Tetrastigma hemsleyanum* can prevent LPS-induced ALI in mice by regulating the TLR4/NF-κB pathway [46]; flavonoids from the fiber roots of *Sanyeqing* alleviate ALI by inhibiting the NLRP3/ferroptosis pathway through a dual pathway [47]; flavonoids from *Abutilon theophrasti* leaves protect against LPS-induced ALI in mice through the NF-κB and MAPK signaling pathways [48]; and Physalin B improves the inflammatory response in mice with LPS-induced ALI by activating the PI3K/Akt pathway and inhibiting NF-κB and NLRP3 [49].

This study explores the protective effects of flavonoid-rich extract from *Bombyx batryticatus* (FBB) against LPS-induced ALI and elucidates its molecular pathways. We employed an integrated approach, utilizing advanced UPLC-MS/MS analysis, network pharmacology, and experimental validation in both cell and animal models. This approach aimed to identify the bioactive components of FBB, clarify its multi-target mechanism of action, and provide a pharmacological foundation for developing new FBB-based treatments for ALI.

## 2. Results

### 2.1. Purification and Compositional Analysis of FBB

The flavonoid content was quantified both before and after purification using a rutin-equivalent standard curve (Appendix A). Purification through AB-8 macroporous resin chromatography, employing a stepwise ethanol–water gradient, significantly increased the flavonoid content from 4.5 to 55.1 mg/g, effectively enriching the target compounds. Ultra Performance Liquid Chromatography–Tandem Mass Spectrometry (UPLC-MS/MS) analysis revealed high chemical consistency between the purified and crude extracts, with a Pearson correlation coefficient of 0.95 (Appendix A), confirming the preservation of the principal flavonoid profile. Using UPLC-MS/MS and a customized flavonoid database, we identified 163 flavonoid compounds that met Level 2 confidence standards according to the COSMOS Metabolomics Standards Initiative, based on accurate mass and MS/MS fragmentation patterns. These flavonoids included 10 chalcones, 17 flavanones, 6 flavanonols, 49 flavones, 61 flavonols, 2 isoflavones, 17 anthrones, and 1 flavanol (Appendix A; Appendix A). This comprehensive profile provides a chemical foundation for investigating FBB’s multi-component, multi-target mechanisms in future pharmacological studies.

### 2.2. Network Pharmacology Analysis of FBB’s Effects on Lung Injury

Database mining identified 569 potential FBB targets and 3459 genes associated with ALI. A comparative analysis revealed 273 overlapping targets, suggesting their potential as therapeutic targets for FBB in lung injury (Figure 1A). To identify core bioactive constituents, we constructed a detailed compound-target interaction network. Network topology analysis showed that 30 flavonoid compounds had the highest degree values, a crucial network parameter, warranting further investigation (Figure 1B, Appendix A).

The protein–protein interaction (PPI) network was constructed using the STRING database (https://string-db.org/) with a high confidence interaction score threshold of >0.9, highlighting significant interconnectivity among potential FBB targets (Figure 1C). Functional enrichment analysis identified the top 10 significantly enriched terms (*p* < 0.01, FDR < 0.05). As depicted in Figure 1D, Gene Ontology (GO) analysis showed a strong association with inflammatory response and oxidative stress response, both ranking among the top biological processes. Kyoto Encyclopedia of Genes and Genomes (KEGG) pathway enrichment analysis demonstrated significant involvement in signal transduction cascades linked to ALI pathogenesis, particularly the MAPK signaling pathway. Through systematic evaluation of network topology parameters, 38 core target proteins were identified (Appendix A). These proteins are primarily involved in several biologically significant pathways: PI3K-AKT signaling (*PIK3R1*, *PIK3CA*, *PIK3CD*, *AKT1*, and *IGF1R*), MAPK signaling (*ERK2*, *ERK1*, *JNK1*, *JNK2*, *p38α*, and *HRAS*), JAK-STAT signaling (*STAT3*, *STAT1*, and *JAK2*), apoptosis-related pathways (*TP53*, *BCL2*, *CASP3*, and *MDM2*), growth factor receptor signaling (*EGFR*, *ERBB2*, *PDGFRA*, *PDGFRB*, and *KDR*/*VEGFR2*), and inflammation/immunomodulation (*TNF*, *PTGS2*/*COX-2*, *HIF1A*, and *NCOR*2). Notably, *TNF* acts as a core upstream activator of the NF-κB pathway, *PTGS2*/*COX-*2 serves as a classic downstream effector of NF-κB, and *AKT* activates NF-κB by phosphorylating IKKα/β.

### 2.3. FBB Mitigates Pro-Inflammatory Cytokines in LPS-Triggered ALI

The anti-inflammatory effects of FBB were evaluated using an LPS-induced murine model. The Enzyme-Linked Immunosorbent Assay (ELISA) showed that LPS significantly increased proinflammatory cytokine levels in both serum and bronchoalveolar lavage fluid (BALF) compared to controls. Pretreatment with FBB reduced these elevations in a dose-dependent manner. In serum, high-dose FBB (160 mg/kg) decreased IL-1β, IL-6, and TNF-α levels to 77.11, 113.45, and 854.68 pg/mL, respectively, representing reductions of 22%, 26%, and 23% from LPS-induced levels (all *p* < 0.001). These reductions were greater than those achieved by DEX (5 mg/kg), which lowered levels by 20%, 16%, and 14% (*p* < 0.05 for IL-6 and TNF-α comparisons between FBB 160 mg/kg and DEX). Low-dose FBB (40 mg/kg) also significantly inhibited some cytokine levels (*p* < 0.05). Similar response patterns were observed in BALF samples (Figure 2).

The qRT-PCR results confirmed FBB’s anti-inflammatory effects at the gene expression level. The LPS challenge significantly increased mRNA expression of *IL-1B*, *IL-6*, and *TNF-α* by 51.36-fold, 10.14-fold, and 78.45-fold, respectively, compared to the control (all *p* < 0.001). Pretreatment with DEX reduced these expressions by approximately 72.6%, 63.7%, and 72.7%, lowering the expression values to 14.09-fold, 3.69-fold, and 21.40-fold, respectively (all *p* < 0.01). High-dose FBB pretreatment showed a stronger inhibitory effect, decreasing expression by about 84.6%, 81.9%, and 83.0%, resulting in expression values of 7.90-fold, 1.83-fold, and 13.32-fold (all *p* < 0.001). Low-dose FBB (40 mg/kg) also significantly reduced expression, though less effectively than the high dose, with reductions of approximately 63.1%, 44.1%, and 46.7% (*p* < 0.05).

### 2.4. FBB Reduced iNOS and COX-2 Levels in LPS-Induced ALI Mice

The qRT-PCR analysis revealed that LPS exposure significantly increased pulmonary transcript levels of *iNOS* (4.32-fold; *p* < 0.001) and *COX-2* (11.26-fold; *p* < 0.001) compared to the control group, as illustrated in Figure 3A,B. Both FBB doses (40 and 160 mg/kg) reduced these increases in a dose-dependent manner. Specifically, the high-dose FBB (160 mg/kg) decreased *iNOS* expression to 33.6% of the LPS group levels (*p* < 0.001), showing greater efficacy than the 40 mg/kg dose, which reduced it to 53.1% (*p* < 0.01). Similarly, *COX-2* expression was reduced to 21.0% (*p* < 0.001) and 49.7% (*p* < 0.001) by the high and low FBB doses, respectively. Notably, the 160 mg/kg FBB restored *iNOS* expression to 1.45-fold of basal levels, demonstrating similar effectiveness to the 5 mg/kg DEX positive control (1.38-fold; *p* > 0.05). All mRNA levels were normalized to *GAPDH*.

Western blot analysis confirmed that LPS stimulation significantly increased iNOS and COX-2 protein expression by 44.3-fold and 8.3-fold, respectively (*p* < 0.001) (Figure 3C–E). Following FBB intervention, the 160 mg/kg group markedly inhibited iNOS expression, reducing it to 21.7% (*p* < 0.001) of the LPS group level, and COX-2 expression to 16.1% (*p* < 0.001), nearly reaching the basal level at 1.33 times (*p* ≥ 0.05). In contrast, the 40 mg/kg group demonstrated weaker inhibition, with iNOS reduced to 56.0% (*p* < 0.05) and COX-2 to 37.1% (*p* < 0.01). It is noteworthy that 160 mg/kg FBB showed similar inhibitory effects on iNOS and COX-2 as DEX (*p* ≥ 0.05).

### 2.5. FBB Alleviates LPS-Induced Lung Injury in Mice

Microscopic observation revealed that the overall structure of the lung tissue in the control group remained essentially normal. Alveolar epithelial cells arranged precisely and orderly, with no abnormalities such as edema or necrosis. Interstitial blood vessels showed no significant congestion or dilation. The bronchial wall thickness, indicated by the blue arrows, was normal, mucus secretion was adequate, and no significant infiltration of inflammatory cells occurred (Figure 4A). The Smith score for this group was 0. In contrast, the lung tissue in the LPS-treated group exhibited significant pathological changes. The alveolar structure sustained severe damage, with extensive areas of collapse, and pathological changes such as bullae and epithelial cell necrosis were evident. Patchy hemorrhages appeared in the interstitial tissue, as indicated by the black arrows. The bronchial wall, also indicated by the blue arrows, thickened significantly, with numerous inflammatory cells infiltrating the lumina. Additionally, the mucus layer within the trachea thickened significantly (Figure 4A). The Smith score for this group was 4. The overall severity of lung tissue lesions was significantly improved in the groups treated with 160 mg/kg FBB and DEX. Alveolar collapse and bullae were observed in localized areas. Slight congestion occurred in the interstitial tissue, as indicated by the black arrow in the figure (Figure 4A). The bronchial wall, marked by the blue arrow, exhibited some thickening, accompanied by a minor infiltration of inflammatory cells. Additionally, the amount of mucus in the trachea increased. The Smith score for both groups was 1. In contrast, the lung tissue lesions in the group treated with 40 mg/kg FBB remained more severe. The alveolar structure collapsed, resulting in the formation of large bullae. Severe patchy hemorrhages were present in the interstitial tissue, as indicated by the black arrow in the figure. The bronchial wall, also indicated by the blue arrow, showed thickening, accompanied by a substantial infiltration of inflammatory cells and a significant increase in mucus in the trachea. The Smith score for this group was 3. The Evans Blue extravasation assay demonstrated that the LPS group exhibited a 2.6-fold increase in dye leakage compared to the control group (*p* < 0.001). High-dose FBB intervention reduced extravasation by 50.1% (*p* < 0.001), while low-dose FBB decreased it by 16.9% (*p* < 0.05; Figure 4B). LPS stimulation significantly increased the lung wet/dry (W/D) ratio, an indicator of tissue water content, by 14.3% (*p* < 0.001). FBB pretreatment dose-dependently lowered this ratio, achieving reductions of 9.2% and 13.1% relative to the LPS group for the 40 mg/kg (*p* < 0.01) and 160 mg/kg (*p* < 0.001) doses, respectively (Figure 4C). Furthermore, FBB treatment significantly attenuated LPS-induced increases in total protein concentrations in BALF, with reductions of 44.1% in the 160 mg/kg group and 22.4% in the 40 mg/kg group (all *p* < 0.001; Figure 4D). Collectively, these findings indicated that FBB effectively ameliorated LPS-induced pulmonary vascular hyperpermeability, potentially through mechanisms that preserved endothelial barrier integrity and suppressed plasma protein extravasation.

### 2.6. FBB Exhibited No Significant Effect on RAW264.7 Cell Growth

In this study, the RAW264.7 murine macrophage cell line was used as the in vitro model. Cell proliferation exhibited concentration-dependent inhibition within the 0.1–10 mg/L LPS concentration range. Specifically, treatments with 0.1 mg/L and 1 mg/L LPS did not significantly affect cellular viability (*p* > 0.05). However, 5 mg/L LPS significantly reduced proliferation to 77.21% (*p* < 0.05), with an inhibition rate of 22.8%, and 10 mg/L LPS further decreased proliferation to 72.70% (*p* < 0.05), corresponding to an inhibition rate of 27.3% (Appendix A). Consequently, 1 mg/L LPS was selected for subsequent inflammation modeling. Following 24 h of treatment with 10–160 mg/L FBB, RAW264.7 cells maintained viability between 99.67% and 102.71% (compared to 100% in the control group; *p* ≥ 0.05), with no statistically significant differences compared to the control group, indicating FBB’s biosafety within this concentration range (Appendix A). Under the 1 mg/L LPS-induced inflammation model, the cell survival rates for the 10–160 mg/L FBB co-treatment groups ranged from 97.16% to 100.77% (*p* ≥ 0.05). The positive control DEX group showed a survival rate of 100.97%, while the LPS-only group exhibited a rate of 95.04%, with no significant differences among all groups (Appendix A). Collectively, these results demonstrate that FBB did not cause significant cytotoxicity at concentrations up to 160 mg/L, whether used alone or in combination with LPS.

### 2.7. FBB Exhibits Anti-Apoptotic Effects Against LPS-Induced Cell Death

LPS significantly induced apoptosis in RAW264.7 cells compared to the control group (*p* < 0.05). Consistent with network pharmacology predictions, FBB treatment showed a concentration-dependent anti-apoptotic effect. As shown in Figure 5, fluorescence microscopy with Hoechst 33342/PI double-staining revealed that high concentrations of FBB (160 and 40 mg/L) significantly reduced LPS-induced apoptosis, as indicated by fewer cells with nuclear condensation and fragmentation (*p* < 0.001 and *p* < 0.05, respectively). In contrast, a low concentration of FBB (10 mg/L) did not show a statistically significant effect (*p* > 0.05). These findings confirm the anti-apoptotic activity of FBB at higher concentrations.

### 2.8. FBB Suppresses LPS-Induced Pro-Inflammatory Cytokine Release in RAW264.7 Cells

ELISA and qRT-PCR were employed to quantitatively assess the expression and release of pro-inflammatory factors (IL-1β, IL-6, and TNF-α) in RAW264.7 cells. ELISA results (Figure 6A–C) revealed that exposure to 1 mg/L LPS significantly increased the secretion of these cytokines in RAW264.7 murine macrophages compared to control levels: IL-1β rose by 126.7%, IL-6 by 517.2%, and TNF-α by 435.0% (all *p* < 0.001). Various concentrations of FBB demonstrated a dose-dependent inhibitory effect. At 160 mg/L, FBB reduced IL-1β to 51.7% of the LPS group’s level (restoring it to the basal level), IL-6 to 70.6%, and TNF-α to 70.5% (all *p* < 0.001). Notably, FBB’s inhibition of IL-6 and TNF-α surpassed that of the positive control drug DEX (*p* < 0.05). At 40 mg/L, FBB significantly inhibited IL-1β, IL-6 and TNF-α, although its effect was weaker than that of the high-dose group. At 10 mg/L, FBB only reduced IL-6 and TNF-α, with no impact on IL-1β.

The results of qRT-PCR experiments showed that 160 mg/L FBB could significantly inhibit the gene expression of inflammatory factors induced by LPS, reducing the levels of *IL-1β*, *IL-6*, and *TNF-α* to only 26.7%, 7.1%, and 8.9% of those in the LPS group, respectively (*p* < 0.001) (Figure 6D–F). Notably, the inhibitory effect of 160 mg/L FBB on *IL-6* (inhibition rate of 92.9%) was significantly higher than that of the positive control drug DEX (inhibition rate of 57.5%) (*p* < 0.05); the inhibitory effect on *TNF-α* (inhibition rate of 91.1%) was also significantly higher than that of DEX (inhibition rate of 51.1%) (*p* < 0.01). At a concentration of 40 mg/L, FBB could still effectively inhibit the expression of *IL-6* and *TNF-α*, restoring them to 54.7% (*p* < 0.01) and 43.7% (*p* < 0.001) of the levels in the LPS group, respectively. However, its inhibitory effect on *IL-1β* was weak, only reducing it to 77.2% of the LPS group (*p* ≥ 0.05). At 10 mg/L, FBB did not significantly inhibit *IL-1β* and *IL-6* (*p* ≥ 0.05), only reducing *TNF-α* to 60.4% of the LPS group (*p* < 0.01).

### 2.9. FBB Modulates the Expression of Inflammatory Cytokines and Receptor-Related Genes in RAW264.7 Cells

The study evaluated the impact of FBB on inflammatory cytokine and receptor-related gene expression in LPS-stimulated RAW264.7 cells using a qRT-PCR array. As shown in Figure 7, LPS markedly increased the expression of multiple inflammatory genes in these cells. Co-treatment with FBB significantly countered these LPS-induced changes. Notably, the FBB + LPS combination substantially reduced key proinflammatory cytokines, including II-6, II-1α, and II-1β (Figure 7A); chemokines Ccl3 and Ccl4 (Figure 7B); and other mediators such as Csf2 and Tnf (Figure 7C), with effects comparable to or exceeding those of DEX. Unlike DEX’s broadly suppressive action, FBB selectively upregulated certain genes after LPS exposure, for example, the decoy receptor II-1r2 (Figure 7A) and chemokine receptors Ccr1 and Ccr9 (Figure 7B). In addition, FBB treatment alone increased the basal expression of genes such as II-4 (Figure 7A) and Ccr1 (Figure 7B).

### 2.10. FBB Inhibits LPS-Induced NO Production and Attenuates iNOS/COX-2 Expression in RAW264.7 Cells

The LPS challenge significantly activated pro-inflammatory responses in RAW264.7 macrophages, increasing nitric oxide (NO) production by 6.5 times compared to control levels (*p* < 0.01). Additionally, the mRNA expression of inducible *iNOS* and *COX-2* was markedly upregulated, reaching 30.3-fold and 1963.3-fold (*p* < 0.01) of control levels, respectively. Intervention with FBB demonstrated a dose-dependent anti-inflammatory effect. Specifically, FBB at concentrations of 160, 40, and 10 mg/L significantly reduced LPS-induced NO levels to 44.2%, 61.6%, and 61.5% of the LPS group (*p* < 0.01), respectively (Figure 8A). At the gene transcription level, 160 mg/L FBB effectively suppressed *iNOS* and *COX-2* mRNA expression to 9.9% and 2.1% of the LPS group (*p* < 0.01), respectively. Meanwhile, 40 mg/L FBB reduced their expression to 41.5% and to 64.5%, respectively (*p* < 0.01). In contrast, 10 mg/L FBB only reduced iNOS levels to 67.0% but failed to reverse the increase in COX-2 (Figure 8B,C) (*p* ≥ 0.05). Notably, the inhibitory impact of 160 mg/L FBB on *iNOS* mRNA was superior to that of the positive control drug DEX, and the degree of *COX-2* inhibition (98%) was considerably higher than that achieved by DEX (67.8%) (*p* < 0.05).

Western blot analysis confirmed (Figure 8D–F) that both 160 mg/L and 40 mg/L FBB significantly reduced LPS-induced expression of COX-2 and iNOS proteins. Specifically, 160 mg/L FBB decreased COX-2 and iNOS proteins by 75.2% and 71.7% (all *p* < 0.001), respectively, while 40 mg/L FBB reduced them by 55.5% and 46.0% (*p* < 0.01). Notably, the inhibition of COX-2 by 160 mg/L FBB was significantly stronger than that by DEX (*p* < 0.05). In contrast, 10 mg/L FBB did not significantly inhibit iNOS protein expression but still resulted in a 33.8% decrease in COX-2 protein.

### 2.11. FBB Mitigates LPS-Induced Inflammation via the PI3K, MAPK, and NF-κB Pathways

This study aimed to clarify how FBB modulates the LPS-induced inflammatory response in RAW264.7 cells by examining the activation of key inflammatory signaling pathways (PI3K, MAPK, and NF-κB) using Western blotting (WB). The results indicated that FBB, particularly at high concentrations (160 mg/L), effectively inhibited the activation of these pathways by LPS. The PI3K pathway demonstrated that 160 mg/L FBB significantly reversed the strong induction of PI3K (0.88-fold compared to LPS at 3.01-fold, *p* < 0.01) and mTOR (1.81-fold compared to LPS at 11.87-fold, *p* < 0.01) caused by LPS. It also maintained AKT1 activity near the basal level (1.33-fold, *p* ≥ 0.05). Additionally, 40 mg/L FBB partially inhibited PI3K (1.35-fold) and mTOR (4.15-fold) induction (*p* < 0.01), while 10 mg/L FBB was ineffective in inhibiting this pathway (*p* ≥ 0.05) (Figure 9A). In the MAPK pathway, FBB treatment significantly reduced LPS-induced p38 MAPK phosphorylation (160 mg/L: 1.28-fold compared to LPS at 3.42-fold, *p* < 0.01). For the ERK pathway’s key kinases, MEK1/2, 160 mg/L FBB (1.55-fold, *p* < 0.05) effectively diminished the LPS induction effect (2.43-fold), showing a dose-dependent inhibitory effect (Figure 9B). Regarding the NF-κB pathway, FBB significantly inhibited LPS-induced IκB-α degradation, as indicated by the phosphorylation level. The IκB-α level in the 160 mg/L FBB group (1.38-fold) was significantly higher than in the LPS group (6.08-fold), surpassing the inhibitory effect of the positive control drug DEX (Figure 9C).

## 3. Discussion

Flavonoids, a class of naturally abundant polyphenolic compounds, demonstrated significant anti-inflammatory [50,51] and antioxidant properties [45,52]. In this study, we used UPLC-MS/MS and flavonoid monomer database to analyze FBB, identifying a total of 163 flavonoids. We then screened FBB’s target proteins through network pharmacology databases. After integrating the data, we identified 273 target genes and 38 core protein targets associated with FBB’s treatment of ALI. Pathway enrichment analysis revealed that these flavonoids primarily exerted anti-inflammatory effects by modulating apoptosis, protein phosphorylation, cytosol, ATP binding, and kinase activity. Furthermore, database analysis showed that FBB acted through multi-target synergistic effects, systematically intervening in the complex pathological network.

Oxidative stress played a central role in the pathological progression of ALI [53]. During ALI development, polymorphonuclear neutrophils (PMNs) adhered to vascular endothelial (VE) cells, migrated through endothelial cells (ECs) to inflammatory sites, and activated, producing excess oxygen free radicals (OFR) [54,55]. This oxidative damage impaired the regulation of alveolar-capillary barrier permeability, leading to protein-rich pulmonary edema (PE), a critical mechanism in ALI’s pathological cascade [56]. Pathogenic factors like LPS triggered mitochondrial-dependent ROS generation in phagocytes by activating the NADPH oxidase (NOX) complex and disrupting electron transport chain function. This induced excessive ROS production, perpetuating continuous oxidative damage [57]. The study investigated FBB’s regulatory effects on LPS-induced oxidative stress. In vivo results demonstrated that FBB significantly suppressed iNOS and COX-2 expression at both mRNA and protein levels in lung tissue. In vitro, FBB dose-dependently inhibited these inflammatory mediators in RAW264.7 macrophages, reducing NO levels in cell culture supernatants.

The core pathology of ALI involved uncontrolled pulmonary inflammation and an imbalance between pro- and anti-inflammatory systems. Notably, chemokine-driven immune cell infiltration directly contributed to the disruption of the alveolar epithelial barrier [28]. Our study showed that in LPS-induced ALI mouse models, pro-inflammatory mediators such as IL-1β, IL-6, and TNF-α significantly increased in BALF, peripheral blood, and cell culture systems. The administration of FBB effectively normalized these elevated expression levels. This regulatory effect indicated that FBB exerted therapeutic effects by targeting key nodes in the inflammatory cascade.

Using qRT-PCR and Western blot analyses, this study showed that FBB modulated LPS-induced inflammation in RAW264.7 macrophages via a multi-target immunoregulatory mechanism distinct from the broad immunosuppression produced by DEX. FBB markedly suppressed phosphorylation of key proteins in the PI3K/AKT, MAPK, and NF-κB pathways. Consistent with NF-κB inhibition, FBB downregulated proinflammatory cytokines IL-1β, IL-6, and TNF-α, as well as chemokines Ccl3, Ccl22, and Cxcl10. Interestingly, it simultaneously upregulated several chemokine receptors, including Ccr1–Ccr4 and Cxcr4. These findings suggested that FBB exerted cytoprotective and anti-inflammatory effects by synergistically inhibiting multiple inflammatory signaling pathways and remodeling the immune microenvironment, highlighting its potential therapeutic relevance in acute lung injury (ALI). The upregulation of chemokine receptors by FBB likely indicated its ability to resolve inflammation through mechanisms different from traditional immunosuppression. Unlike DEX, which broadly suppressed immune activity by directly inhibiting NF-κB and AP-1 via glucocorticoid receptors, FBB modulated chemokine-mediated cell migration and macrophage polarization. This modulation possibly favored an M2-like phenotype or the recruitment of regulatory immune cells. Furthermore, FBB’s inhibition of PI3K/AKT and MAPK signaling might have indirectly adjusted receptor expression through feedback mechanisms. Remarkably, FBB alone increased baseline receptor levels without triggering inflammation, suggesting it preconditioned the immune regulatory network. In the LPS model, FBB outperformed DEX in reducing IL-1β and IL-6 levels and selectively regulated chemokines such as Ccl2 and Ccl5. This supported a model where FBB resolved inflammation through immune reprogramming rather than broad suppression, aiding tissue repair and the recovery of homeostasis in ALI.

In the pathogenesis of ALI, the NF-κB, MAPK, and PI3K/Akt pathways formed a tightly interconnected signaling network that collectively regulated inflammatory dysregulation, cell fate decisions, and barrier dysfunction [28]. Upon LPS exposure, these pathways activated simultaneously and engaged in extensive crosstalk. NF-κB acted as the inflammatory master switch [14,15], while MAPK components (p38, JNK) modulated its activation [13,58], and PI3K/Akt enhanced IKKα phosphorylation to amplify pro-inflammatory transcription [59,60]. Concurrently, MAPK promoted apoptosis through ASK1 activation [61] and disrupted barrier integrity via RhoA/ROCK signaling [62]. Meanwhile, PI3K/Akt/mTOR overactivation suppressed autophagy and compromised endothelial junctions [63]. These pathways formed positive feedback loops that sustained inflammatory amplification [64]. In this study, FBB demonstrated synergistic anti-inflammatory activity by concurrently suppressing phosphorylation of p38 and ERK in the MAPK pathway, mTOR/PI3K/Akt in the PI3K cascade, and IκB-α in the NF-κB pathway.

Among the 273 potential targets identified through network pharmacology, functional enrichment showed significant links with the MAPK, PI3K/AKT, apoptosis, and immune-inflammatory signaling pathways. To clarify FBB’s multi-target mechanism, we validated its effects on key network-predicted components such as PI3K, AKT1, TNF, CXCR2, IκB, and ERK1. Our findings demonstrated that FBB simultaneously attenuated the PI3K/AKT, MAPK, and NF-κB pathways, resulting in a synergistic suppression of downstream inflammatory mediators. By modulating various chemokines and cytokines, including the Ccl and Il families, FBB reduced inflammatory network activation, apoptosis, and lung tissue injury, thus providing a multi-target protective effect in ALI. Despite these findings, several limitations warrant attention. First, while UPLC-MS/MS identified 163 flavonoid constituents in FBB, absolute quantification was constrained by the availability of reference standards. Additionally, the bioavailability and metabolic profiles of these compounds have yet to be characterized. Second, experimental validation concentrated on a few central pathways, such as NF-κB and Nrf2/ARE, leaving other network-predicted targets unexplored. Another crucial point is the interspecies variation in pharmacokinetics when translating findings from mice to human pharmacokinetic models. This variation primarily arises from inherent physiological, anatomical, and molecular differences between species, resulting in significant discrepancies in drug absorption, distribution, metabolism, and excretion. Although quantitative extrapolation can be achieved through physiologically based pharmacokinetic modeling and allometric scaling, uncertainties persist due to the interactions of factors such as drug-specific target affinity, conserved metabolic pathways, and substantial phylogenetic distances. Therefore, mouse data must be interpreted with caution, and multiple validations should be conducted alongside supplementary studies utilizing in vitro human systems or organoids [65]. Lastly, since only an LPS-induced ALI model was used, further studies are necessary to evaluate FBB’s efficacy across ALI caused by other factors like sepsis and trauma. Addressing these limitations will advance the modernization of traditional medicine and offer new insights into natural product-based therapies for ALI.

## 4. Materials and Methods

### 4.1. Experimental Materials

*Bombyx batryticatus* was obtained from the Second Affiliated Hospital of Zunyi Medical University (Zunyi, China). Male BALB/c mice (SPF grade, 6–8 weeks old, 20–25 g) were supplied by Zunyi Medical University’s Laboratory Animal Center and housed individually in standard cages (320 × 180 × 160 mm) under controlled conditions (12-h light/dark cycle, 23–25 °C, 45–65% relative humidity) with ad libitum access to food and water. After a 1-week acclimation period, experiments began in accordance with ARRIVE guidelines. All procedures were approved by Zunyi Medical University’s Institutional Animal Care and Use Committee (Approval No. ZMU21-2412-012) and followed The China Council on Animal Care guidelines.

### 4.2. Extraction and Purification of Total FBB

FBB was extracted following previously described protocols [66,67,68]. Briefly, *B. batryticatus* powder was ultrasonically extracted with 63.2% ethanol (solid–liquid ratio, 1:32.2) at 49.5 °C for 40.3 min. After filtration and volume adjustment, the crude extract was obtained. Total flavonoid content was determined by the NaNO_2_–Al(NO_3_)_3_–NaOH colorimetric method using rutin (Aladdin Scientific, Shanghai, China) as the standard. The FBB crude extract was concentrated by rotary evaporation and redissolved in 95% ethanol. Polyamide resin (2 g dry resin per 10 mL extract) was added, and the mixture was loaded onto a column. The column was eluted with pure water until the effluent was colorless, then transferred to an AB-8 resin column packed at three times the mass of *B. batryticatus* and eluted with 50% ethanol. Seven bed volumes (BV) of eluate were collected and evaporated in a water bath to dryness to yield a total flavonoid-enriched product.

### 4.3. UPLC-MS/MS Analysis of FBB Composition

The chemical components of FBB were analyzed by UPLC-MS/MS on an ExionLC™ AD system (SCIEX, Shanghai, China). Separation was performed on an Agilent SB-C18 (Agilent Technologies, Beijing, China) reversed-phase column (2.1 × 100 mm, 1.8 μm). The mobile phase consisted of solvent A (0.1% formic acid in water) and solvent B (0.1% formic acid in acetonitrile). A gradient program was applied: 5% B was linearly increased to 95% over 9 min, held for 1 min, returned to 5% B within 1.1 min, and equilibrated for 2.9 min. All separations used a constant flow rate of 350 μL/min, a column temperature of 40 °C, and a 2 μL injection volume.

Mass spectrometric analysis was performed using electrospray ionization (ESI) with the following optimized parameters: the ionization source was maintained at 550 °C; voltages were set to 5500 V (positive mode) and −4500 V (negative mode); and the nebulizing gases (GS1, GS2, and curtain gas) were regulated at 50, 60, and 25 psi, respectively. The collision cell was operated in high-energy collision-induced dissociation mode. Multiple reaction monitoring (MRM) used nitrogen as the collision gas at medium pressure. For each analyte, specific MRM transitions were individually optimized for declustering potential and collision energy. Metabolite-specific transitions were monitored continuously according to their chromatographic elution profiles.

### 4.4. Network Pharmacology Analysis of FBB in ALI

Structural data for the compounds in FBB were sourced from PubChem and analyzed using the Swiss Target Prediction platform (http://www.swisstargetprediction.ch/ (accessed on 8 December 2025)) to identify potential targets. To gather known targets associated with lung injury, the Gene Cards (https://www.genecards.org/) and OMIM (https://omim.org/) databases were searched using the term “ALI.” Potential therapeutic targets of FBB for lung injury were identified by finding the intersection between ALI-related targets and FBB compound targets, which was then depicted in a Venn diagram. The core bioactive flavonoids in FBB were identified as the 30 compounds with the highest degree values, determined using Cytoscape 3.10.2. Furthermore, a compound-target network connecting FBB, its constituents, and ALI-related targets was constructed and visualized with the same software. A protein–protein interaction (PPI) network was constructed using the STRING (https://string-db.org/) database, focusing on “Homo sapiens” and applying a high-confidence interaction score cutoff of 0.900. Isolated nodes were removed to finalize the PPI network for intersecting targets. After analyzing network topology, proteins with degree values more than twice the median were identified as core targets. Functional enrichment analysis was performed using the DAVID bioinformatics resource (https://davidbioinformatics.nih.gov/). GO and KEGG pathway terms were ranked by ascending *p*-values, with lower *p*-values indicating greater enrichment significance. The ten most significant terms from the BP, CC, and MF ontologies of GO, along with the top ten KEGG pathways, were selected and depicted in a composite line-bar chart. In this chart, the vertical axis lists the functional terms, the upper horizontal axis and line trace show the gene count, and the lower horizontal axis and bars represent the −log10-transformed *p*-value.

### 4.5. Drug Treatment In Vivo

This study aimed to explore the preventive and early protective effects of FBB against ALI. Clinically, ALI often arises from severe underlying conditions like SARS-CoV-2 infection, with the risk increasing as these conditions deteriorate. Consequently, our experimental approach, which involved administering FBB before the LPS challenge, was intended to replicate a scenario where preventive intervention occurs before or at the onset of a potential ALI trigger, such as an aggravated infection.

The Pharmacopoeia of the People’s Republic of China states that the clinical dosage of *B. batryticatus* is about 5–10 g. However, clinicians usually administer 3–15 g based on the patient’s needs [32]. The conversion formula was: Mouse dose (mg/kg) = [Human dose (g/kg) × 1000] × 70 kg × 0.0026/0.02 kg, where 0.0026 is the human-to-mouse equivalent dose coefficient, and 70 kg and 0.02 kg are the estimated human and mouse body weights, respectively. The resulting value was then divided by the ratio of flavonoid concentration increase before and after purification, yielding a mouse dosage of approximately 40–160 mg/kg. LPS and DEX were sourced from Sigma-Aldrich (St. Louis, MO, USA). An ALI model was induced by intratracheal LPS instillation (5 mg/kg; [69]). Forty mice were randomly assigned to five groups (n = 8 each): (1) Control (saline), (2) LPS (5 mg/kg), (3) DEX (5 mg/kg DEX + LPS [70]), (4) FBB-L (40 mg/kg FBB + LPS), and (5) FBB-H (160 mg/kg FBB + LPS). Pretreatments with FBB, DEX, or saline were administered intraperitoneally 1 h before modeling. Subsequently, under anesthesia, the mice received intratracheal LPS treatment, while the control group received a saline injection. Peripheral blood, BALF, and lung tissues were collected 24 h after modeling [71].

### 4.6. Measurement of Lung Wet/Dry Weight Ratio (W/D)

The severity of pulmonary edema was evaluated by the wet-to-dry weight ratio (W/D ratio). Fresh lung specimens were gently blotted to remove surface moisture, immediately weighed to record wet mass, and then dehydrated at 70 °C for 48 h to obtain dry mass.

### 4.7. Histopathological Examination

Fresh lung tissues were fixed for 24 h, trimmed in a fume hood, and placed into labeled dehydration cassettes. The samples underwent dehydration using an automated graded series of ethanol, xylene, and paraffin. Subsequently, the wax-infiltrated tissues were embedded, oriented, and solidified into blocks. Sections, each 3 μm thick, were cut with a rotary microtome, floated on a 40 °C water bath, and transferred to slides for drying at 60 °C. Deparaffinization was conducted using xylene, followed by rehydration through a descending ethanol series and rinsing with tap water. The sections were stained with hematoxylin for 3–5 min, differentiated in 1% acid alcohol, blued in Scott’s tap water substitute, and counterstained with eosin for 5 min. Finally, the sections were dehydrated, cleared in xylene, and mounted with neutral synthetic resin. Lung injury was assessed by observers blinded to the groups using light microscopy. The Smith scoring system was employed, evaluating four parameters: pulmonary edema, alveolar and interstitial inflammation, hemorrhage, and atelectasis. Each parameter received a score from 0 (no injury) to 4 (severe injury, >75% involvement) [72], with the total score being the sum of all four parameters.

### 4.8. Assessment of Pulmonary Capillary Permeability

At 22 h after modeling, mice received an intravenous injection of 1% Evans Blue (Aladdin Scientific, Shanghai, China). At 2 h post-injection, the mice were euthanized, and their lungs were perfused through the left ventricle until the effluent turned colorless. Lung tissue homogenates, prepared with 1 mL of formamide per 100 mg of tissue, were incubated at 37 °C for 24 h. Subsequently, the samples were centrifuged at 12,000× *g* rpm for 30 min. Evans Blue extravasation was quantified by measuring the absorbance of the supernatant at 620 nm.

### 4.9. BALF Collection and Analysis

Twenty-four hours after LPS administration, mice were euthanized. The thoracic cavity and neck were surgically exposed to access the trachea for BALF collection. Lavage was performed three consecutive times with ice-cold PBS (0.5 mL per wash, 4 °C), yielding a combined volume of 1.5 mL. Bronchoalveolar lavage fluid was centrifuged at 3000 rpm and 4 °C for 10 min. The supernatants were aliquoted for measurement of inflammatory cytokines and total protein. Total protein concentration in BALF was determined with a BCA assay kit (Epizyme Biotech, Shanghai, China) following the manufacturer’s instructions.

### 4.10. Blood Collection

Mice were anesthetized with 1.25% avertin before blood was collected through orbital puncture. The blood droplets were directly collected into sterile microcentrifuge tubes and allowed to clot for 1 h at 25 °C. After coagulation, the samples were centrifuged at 4000 rpm for 20 min at 4 °C. The clarified serum supernatant was then aliquoted into fresh 1.5 mL microcentrifuge tubes using precision pipetting for subsequent analytical procedures.

### 4.11. Cell Viability Assay

RAW264.7 cells (Procell system, CL-0190) at 5 × 10^4^/mL were treated for 24 h [71] with varying concentrations of FBB (10–160 mg/L), LPS (0.1–10 mg/L), or DEX (10 mg/L; [69]). After treatment, Cell Counting Kit-8 (CCK-8) reagent (Beyotime, Shanghai, China) was added, and plates were incubated for 2 h. Optical density was then measured at 450 nm.

### 4.12. ELISA

Pro-inflammatory mediator concentrations were measured using ELISA kits from Enzyme-linked Biotechnology, Shanghai, China. For peripheral blood and BALF analysis, standards and diluted samples (1:5) were placed into microplates at 50 μL per well and incubated at 37 °C for 30 min. After washing, an enzyme conjugate was added, and the plate was re-incubated. Color development was initiated with substrates, stopped after 10 min, and absorbance was measured at 450 nm. In cellular experiments, RAW264.7 cells (1 × 10^6^/mL) were treated with PBS, DEX (10 mg/L), or FBB (10–160 mg/L) for 1 h before LPS (1 mg/L) stimulation. Supernatants were collected 24 h after stimulation for quantifying IL-1β, IL-6, and TNF-α.

### 4.13. RNA Isolation and Quantitative Real-Time PCR Analysis

Total RNA was isolated using a commercial extraction kit from Thermo Fisher Scientific, Waltham, MA, USA. Complementary DNA (cDNA) was synthesized with the PrimeScript™ RT reagent system from TaKaRa, Beijing, China. qRT-PCR was conducted using TB Green^®^ Premix Ex Taq™ II and sequence-specific oligonucleotides (Appendix A) on a CFX96 Touch Real-Time PCR platform from Bio-Rad, Hercules, CA, USA. The reaction volume was 25 μL, comprising 12.5 μL of 2× premix master buffer (with Tli RNaseH Plus), 1 μL each of 10 μM forward and reverse primers, 2 μL of cDNA, and 8.5 μL of nuclease-free water. Amplification conditions included an initial denaturation at 95 °C for 30 s, followed by 40 cycles at 95 °C for 5 s and 60 °C for 30 s, as per the manufacturer’s instructions. Gene expression levels were quantified using the comparative threshold cycle (2^−ΔΔCT^) method.

### 4.14. Inflammatory Gene Profiling by qRT-PCR Array

The expression profiles of 90 inflammatory cytokine and receptor genes were analyzed to assess the effects of FBB treatment, alone or in combination with LPS, using a qRT-PCR array kit (wcgene, Shanghai, China; Appendix A). Total mRNA was isolated from RAW264.7 cells assigned to the following experimental groups: control, LPS-treated, DEX + LPS, FBB (160 mg/L) + LPS, and FBB (160 mg/L) alone. Following RNA extraction, cDNA was synthesized by reverse transcription. Subsequent qRT-PCR analysis was performed in accordance with the manufacturer’s protocol.

### 4.15. Western Blot Analysis

Cells and tissue samples were lysed in RIPA buffer supplemented with protease and phosphatase inhibitors (Epizyme Biotech, Shanghai, China) and clarified by centrifugation (12,000× *g*, 15 min, 4 °C). Protein concentrations were measured by BCA assay (Epizyme Biotech, Shanghai, China), and aliquots were adjusted to 2 mg/mL with RIPA and 5× SDS loading buffer before thermal denaturation (100 °C, 10 min). For WB, 30 μg of protein per lane were separated on 10% SDS-PAGE gels and transferred to PVDF membranes (Merck, Darmstadt, Germany). Membranes were blocked in 5% BSA/TBST for 2 h at 25 °C and then incubated overnight at 4 °C with primary antibodies (HuaBio, Hangzhou, China) diluted in TBST. The following primary antibodies were used: inflammatory mediators (iNOS, ER1706-89, 1:2000; COX-2, ET1610-23, 1:2000), phosphorylated signaling proteins (p-P38 MAPK, ET1702-65, 1:2000; p-ERK1/2, ET1601-29, 1:2000; p-AKT1, ET1609-47, 1:2000; p-PI3K, ET1610-36, 1:2000; p-mTOR, ET1608-5, 1:2000; p-IκB-α, ET1603-6, 1:2000), and loading controls (β-actin, HA722023, 1:5000; α-tubulin, ET1705-31, 1:5000; GAPDH, HA721136, 1:5000). After washing with TBST, the membranes were incubated with HRP-conjugated secondary antibodies for 1.5 h at 25 °C. The immunoreactive bands were visualized by chemiluminescence and quantified using Image Lab 6.1 software (Bio-Rad, Hercules, CA, USA), with the signal intensities normalized to those of housekeeping proteins.

### 4.16. Detection of Apoptosis and Necrosis

Cells at a concentration of 1 × 10^5^/mL were pre-treated with FBB (ranging from 10 to 160 mg/L) or DEX (10 mg/L) for 1 h before being stimulated with LPS (1 mg/L). After a 24-h incubation period, nuclear staining was conducted using a Hoechst 33342/PI double-staining kit (Solarbio, Beijing, China) to facilitate fluorescence microscopy evaluation.

### 4.17. NO Content Measurement

Cells (1 × 10^6^ cells/mL; 500 μL per well) were seeded into 24-well plates. After an overnight incubation, drug treatments were administered. Twenty-four hours post-treatment, the supernatants were collected by centrifugation at 12,000× *g* rpm for 5 min at 4 °C. For NO detection, Griess reagent I and II (Beyotime, Shanghai, China) were brought to 25 °C, and standards ranging from 0 to 100 μM were prepared in basal medium. In 96-well plates, 50 μL of each sample or standard was mixed with 50 μL of Griess reagents I and II. Absorbance was then measured at 540 nm.

### 4.18. Data Analysis

Statistical analyses were conducted using GraphPad Prism 10.4 (GraphPad Inc., San Diego, CA, USA). The central tendency and dispersion for each experimental group were expressed as the arithmetic mean ± SEM. For comparisons between multiple groups, one-way ANOVA was used. Results are graphically represented, with statistical significance indicated by asterisks (* *p* < 0.05, ** *p* < 0.01, *** *p* < 0.001, **** *p* < 0.0001).

## 5. Conclusions

This study establishes the chemical and pharmacological foundation of FBB for combating ALI. The purification and characterization process identified 163 flavonoids, underscoring FBB’s multi-component nature. Through integrated network pharmacology and experimental validation, it was demonstrated that FBB protects against ALI via a multi-target mechanism, primarily modulating the PI3K/AKT, MAPK, and NF-κB pathways. As a result, FBB significantly inhibited pro-inflammatory cytokines (IL-1β, IL-6, TNF-α), oxidative stress markers (iNOS, COX-2), and apoptosis. This, in turn, alleviated pulmonary edema, vascular hyperpermeability, and histopathological damage. Notably, FBB showed efficacy comparable to or better than dexamethasone in key parameters without significant cytotoxicity. These findings elucidate the molecular mechanism of FBB and underscore its potential as a multi-target natural candidate for ALI/ARDS treatment.

## Figures and Tables

**Figure 1 ijms-26-12057-f001:**
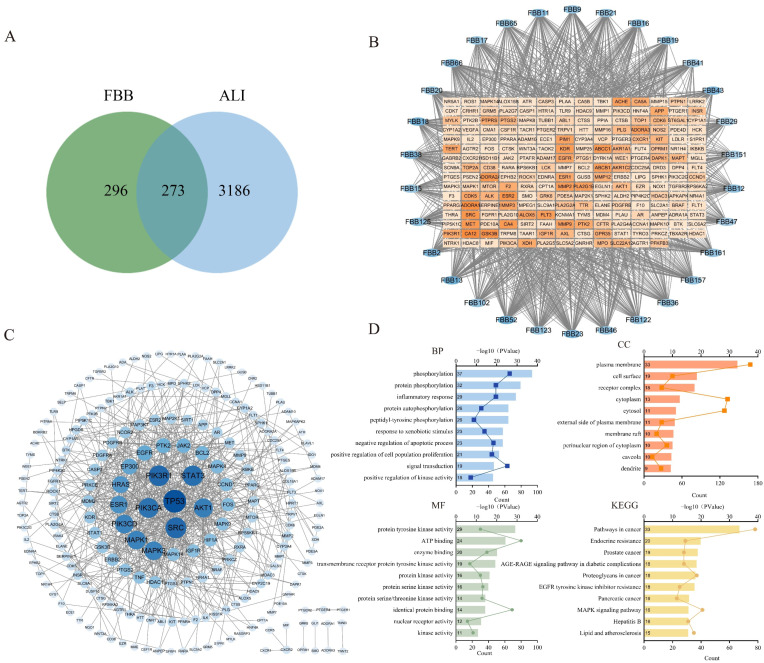
Network pharmacology analysis revealed the multi-target mechanisms of FBB in combating ALI. (**A**) A Venn diagram illustrated the overlap between potential FBB targets (n = 569) and ALI-associated genes (n = 3459), identifying 273 shared targets for further investigation. (**B**) The ‘Compound-Target’ interaction network highlighted 30 core flavonoid components (blue ellipses) with the highest degree values and their related ALI targets (red squares). The darkness of the nodes denoted their degree value. (**C**) Protein–protein interaction analysis of the 273 overlapping targets showed that darker and larger nodes indicated a higher index. (**D**) Functional enrichment analysis of these overlapping targets revealed the top significantly enriched GO terms and KEGG pathways, screened by a *p*-value < 0.05. The bar length represented enrichment significance (−log10(*p*-value)).

**Figure 2 ijms-26-12057-f002:**
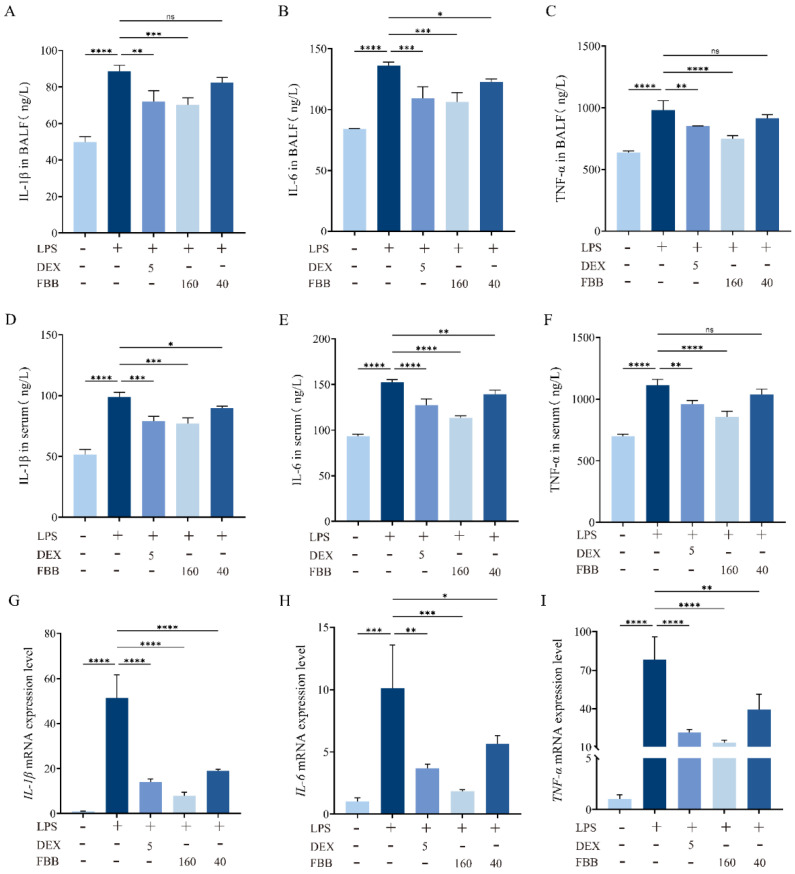
Analysis of inflammatory factors in an LPS-induced murine ALI model. (**A**–**C**) Serum concentrations of IL-1β (**A**), IL-6 (**B**), and TNF-α (**C**) were measured using ELISA. (**D**–**F**) BALF concentrations of IL-1β (**D**), IL-6 (**E**), and TNF-α (**F**) were also quantified by ELISA. (**G**–**I**) The relative mRNA expression levels of IL-1β (**G**), IL-6 (**H**), and TNF-α (**I**) in lung tissue were determined through qRT-PCR. mRNA levels were normalized to *GAPDH* and expressed relative to the control group. Mice received pretreatment with FBB or 5 mg/kg DEX 1 h before LPS challenge (5 mg/kg). Samples were collected 24 h after LPS administration. Data are shown as mean ± SD for n = 3 biologically independent mice per group. Statistical significance was assessed using one-way ANOVA followed by Dunnett’s test. ns, not significant, * *p* < 0.05, ** *p* < 0.01, *** *p* < 0.001, **** *p* < 0.0001.

**Figure 3 ijms-26-12057-f003:**
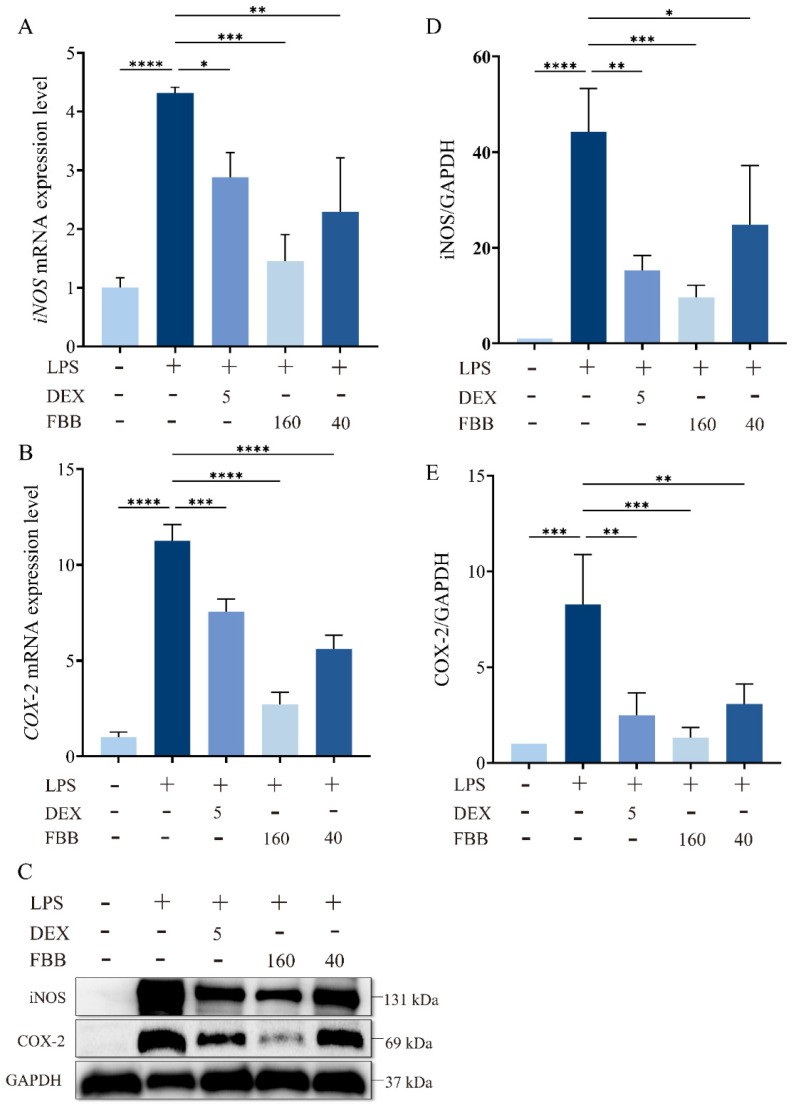
FBB attenuates LPS-induced oxidative stress in murine lung tissue. Mice were pretreated with FBB (40 or 160 mg/kg) or DEX (5 mg/kg) 1 h before LPS (5 mg/kg) administration. Lung tissues were collected 24 h after LPS exposure for analysis. (**A**,**B**) qRT-PCR quantified *iNOS* (**A**) and *COX-2* (**B**) mRNA expression. (**C**–**E**) Western blot assessed iNOS and COX-2 protein expression. Representative blots are shown (**C**), with densitometric quantification of iNOS (**D**) and COX-2 (**E**) protein levels normalized to GAPDH. Data are presented as mean ± SD (n = 3 biologically independent mice per group). Statistical significance was assessed by one-way ANOVA followed by Dunnett’s test. * *p* < 0.05, ** *p* < 0.01, *** *p* < 0.001, **** *p* < 0.0001.

**Figure 4 ijms-26-12057-f004:**
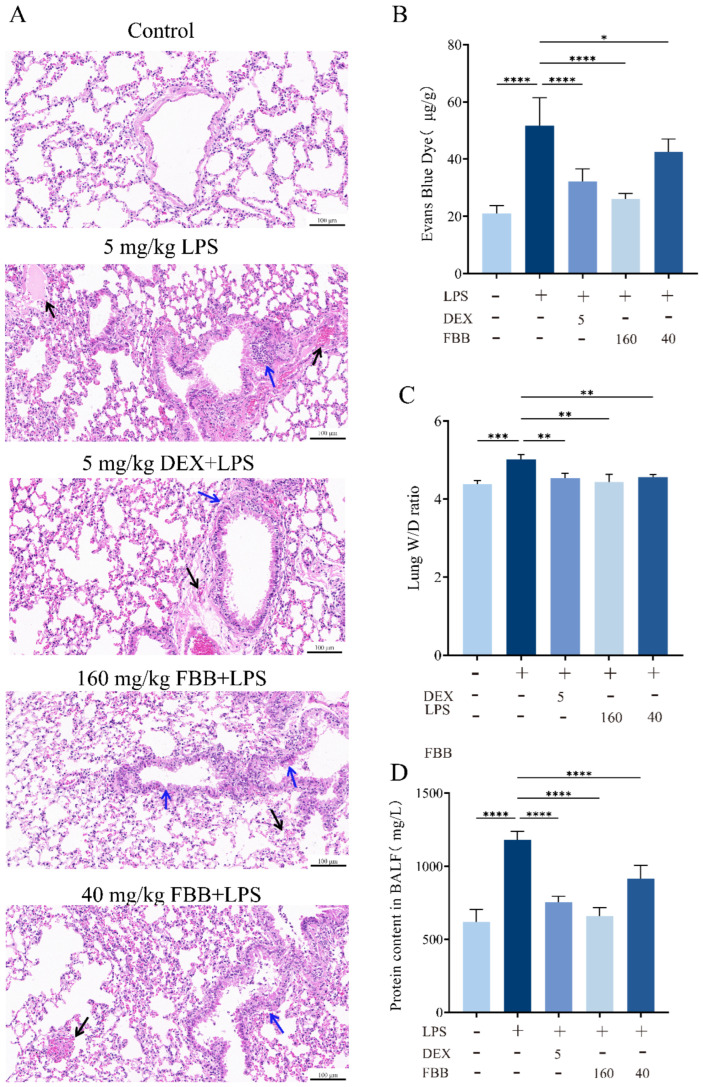
Histopathological and physiological assessment of lung injury in LPS-induced ALI mice. (**A**) Representative photomicrographs of H&E-stained lung tissue (20×). Black arrows indicate interstitial congestion; blue arrows indicate bronchial wall thickening. (**B**) Evans Blue extravasation in lung tissue, quantified 24 h after LPS administration. (**C**) Lung wet-to-dry weight ratio measured 24 h after LPS challenge. (**D**) Total protein concentration in bronchoalveolar lavage fluid. Data are mean ± SD (n = 3 biologically independent mice). Statistical comparisons were performed by one-way ANOVA with Dunnett’s test. * *p* < 0.05, ** *p* < 0.01, *** *p* < 0.001, **** *p* < 0.0001.

**Figure 5 ijms-26-12057-f005:**
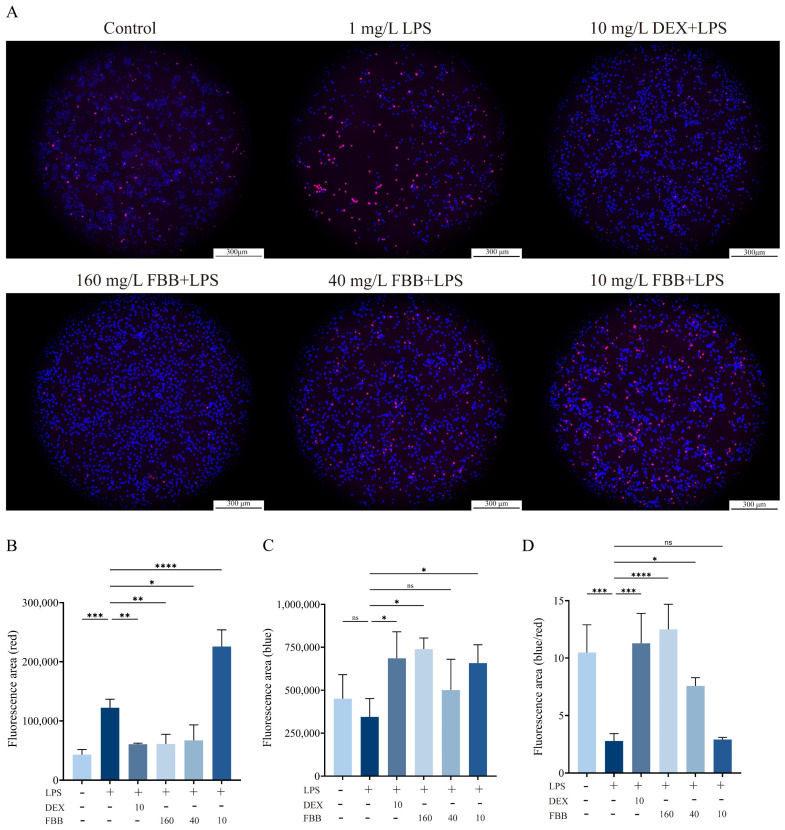
Analysis of apoptosis in LPS-stimulated RAW264.7 cells. (**A**) Representative fluorescence images of cells stained with Hoechst 33342 (blue) and propidium iodide (PI, red) to visualize apoptotic morphology. Viable cells show blue nuclei, while apoptotic cells display condensed or fragmented orange-to-red nuclei (10×). (**B**) Quantification of red fluorescence intensity (PI), indicating late apoptotic and necrotic cells. (**C**) Quantification of blue fluorescence intensity (Hoechst 33342), indicating total cell count and early apoptotic changes. (**D**) Quantitative analysis of apoptosis rate, expressed as the percentage of red fluorescence intensity divided by blue fluorescence intensity. Data are presented as mean ± SD (n = 3 independent experiments). Statistical significance was determined by one-way ANOVA followed by Dunnett’s test. ns, not significant, * *p* < 0.05, ** *p* < 0.01, *** *p* < 0.001, **** *p* < 0.0001.

**Figure 6 ijms-26-12057-f006:**
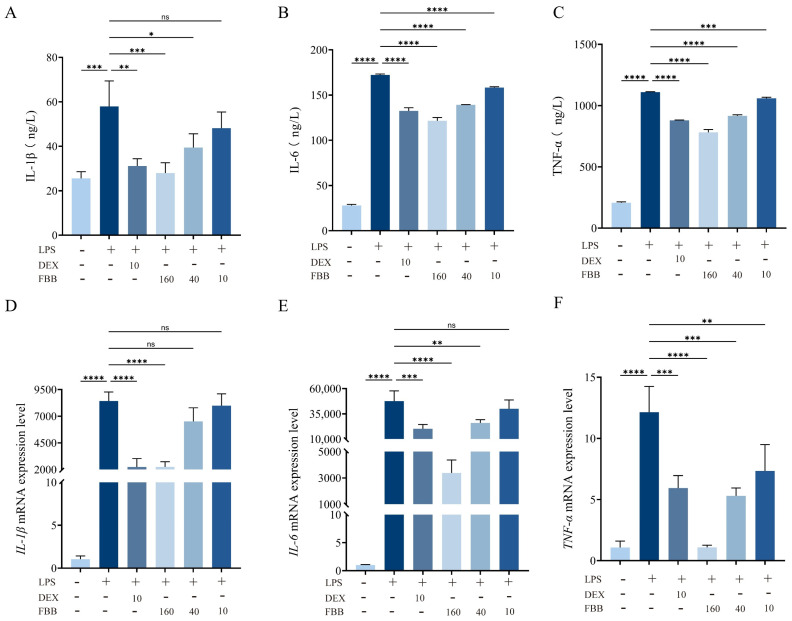
Analysis of inflammatory mediators in LPS-stimulated RAW264.7 cells. (**A**–**C**) Concentrations of IL-1β (**A**), IL-6 (**B**), and TNF-α (**C**) in cell culture supernatant, quantified by ELISA after 24 h of treatment. (**D**–**F**) Relative mRNA expression levels of *IL-1β* (**D**), *IL-6* (**E**), and *TNF-α* (**F**) in cells, determined by qRT-PCR after 24 h of treatment. mRNA levels were normalized to *GAPDH* and are expressed relative to the control group. Data are presented as mean ± SD (n = 3 independent biological replicates). Statistical significance was determined by one-way ANOVA followed by Dunnett’s test. ns, not significant, * *p* < 0.05, ** *p* < 0.01, *** *p* < 0.001, **** *p* < 0.0001.

**Figure 7 ijms-26-12057-f007:**
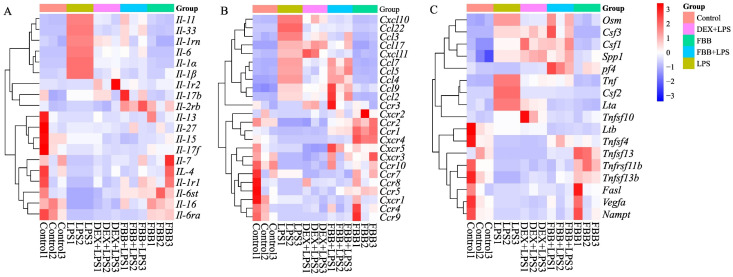
Transcriptional profiling of inflammatory mediators in LPS-stimulated RAW264.7 cells. The heatmap displays the relative expression levels of a panel of inflammatory genes across different treatment groups (from left: Control, LPS (1 mg/L), DEX (10 mg/L) + LPS, FBB (160 mg/L) + LPS, FBB (160 mg/L)). Gene expression was quantified by qRT-PCR and is presented as Z-scores calculated per row (gene). Rows (genes) were clustered using hierarchical clustering with complete linkage. The color key from blue to red represents low to high Z-score values. (**A**) A subset of interleukin and receptor genes. (**B**) A subset of chemokine and receptor genes. (**C**) A subset of other cytokines and receptor genes.

**Figure 8 ijms-26-12057-f008:**
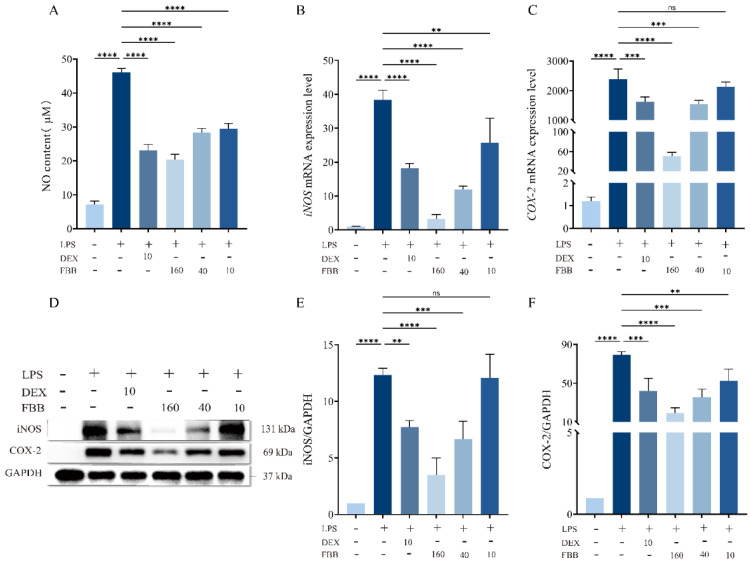
Analysis of oxidative stress-related mediators in LPS-stimulated RAW264.7 cells. (**A**) NO concentration in cell culture supernatant measured by Griess reagent after 24 h of treatment. (**B**,**C**) Relative mRNA expression levels of *iNOS* (**B**) and *COX-2* (**C**) determined by qRT-PCR after 24 h of treatment. mRNA levels were normalized to *GAPDH*. (**D**) Representative immunoblots of iNOS and COX-2 proteins. (**E**,**F**) Quantitative analysis of iNOS (**E**) and COX-2 (**F**) protein expression levels normalized to GAPDH. Data are presented as mean ± SD (n = 3 independent biological replicates). Statistical significance was determined by one-way ANOVA followed by Dunnett’s test. ns, not significant, ** *p* < 0.01, *** *p* < 0.001, **** *p* < 0.0001.

**Figure 9 ijms-26-12057-f009:**
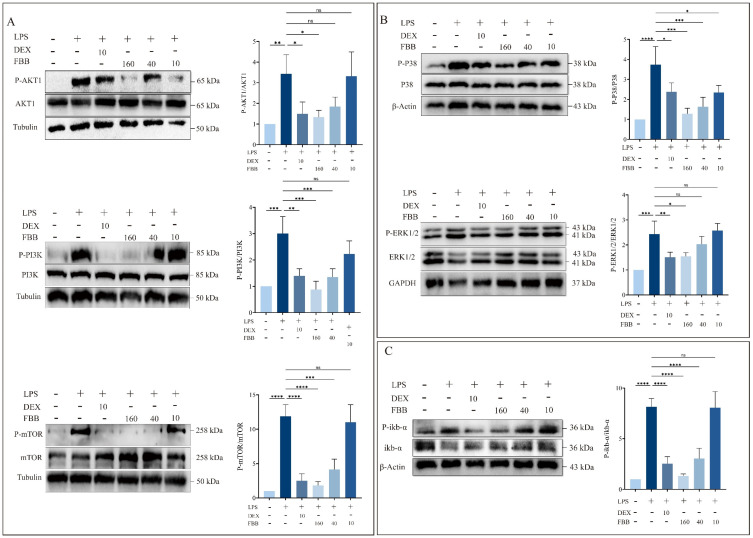
Analysis of key protein phosphorylation in PI3K/AKT, MAPK, and NF-κB signaling pathways in LPS-stimulated RAW264.7 cells. Protein phosphorylation was analyzed by WB. (**A**) Phosphorylation levels of proteins in the PI3K/AKT pathway. Blots were probed for phosphorylated and total forms of AKT, PI3K, and mTOR. (**B**) Phosphorylation levels of proteins in the MAPK pathway. Blots were probed for phosphorylated and total forms of ERK1/2 and P38. (**C**) Phosphorylation level of IkB-α protein in the NF-κB pathway. Blots were probed for phosphorylated and total IkB-α. Quantification graphs represent the ratio of phosphorylated protein to total protein. Data are presented as mean ± SD (n = 3 independent biological replicates). Statistical significance was determined by one-way ANOVA followed by Dunnett’s test. ns, not significant, * *p* < 0.05, ** *p* < 0.01, *** *p* < 0.001, **** *p* < 0.0001.

## Data Availability

The original contributions presented in this study are included in the article/Appendix A. Further inquiries can be directed to the corresponding author.

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
