# Peer review of "Flavonoid-Rich Extract from Bombyx batryticatus Alleviate LPS-Induced Acute Lung Injury via the PI3K/MAPK/NF-κB Pathway"

_ijms, 2025, doi:10.3390/ijms262412057_

Round 1

Reviewer 1 Report (New Reviewer)

Comments and Suggestions for Authors

This study explores the protective effects of flavonoid-rich extracts from Bombyx batryticatus (FBB) against LPS-induced acute lung injury (ALI) through in vivo and in vitro assays. The authors employed UPLC-MS/MS, network pharmacology, and experimental validation to identify 163 flavonoid compounds and demonstrated that FBB mitigates inflammation, oxidative stress, and apoptosis via the PI3K/MAPK/NF-κB signaling pathways. The findings provide mechanistic insights into FBB’s therapeutic potential as a multi-target natural compound against ALI. However, before publication, the manuscript requires substantial revisions in experimental clarity, data quantification, figure interpretation, and integration of recent literature to strengthen scientific rigor.

Comments for the authors

Comment 1. The title accurately reflects the study, but the abstract feels overloaded with methodological details. Revise it to emphasize novelty and the most significant biological insights rather than repeating assay lists. Clearly state what differentiates FBB’s effect from other flavonoid-based treatments for ALI.

Comment 2. The introduction builds a logical background, but it misses discussion of the most recent mechanistic studies on plasma-based antioxidant or flavonoid systems in pulmonary inflammation. Add a paragraph discussing recent molecular perspectives on ALI management, particularly emphasizing multi-target anti-inflammatory strategies (See: IJMS, 2025, 10.3390/ijms26062678).

Comment 3. The literature review is week. I recommend addition of recent studies integrating flavonoid-based natural therapeutics with redox or immune modulation pathways (see: IJMS, 2025, 10.3390/ijms26157381).

Comment 4. Provide quantitative validation for network pharmacology predictions, specifically, which of the predicted 273 targets were experimentally confirmed? Discuss the biological meaning of “multi-target synergy” with quantitative backing.

Comment 5. The data on iNOS and COX-2 suppression are promising, yet the comparison with DEX seems inconsistent, present quantitative ratios with proper normalization to control for assay variability.

Comment 6. Discuss the pharmacological relevance of FBB dosage

Comment 7. Figure shows significant gene expression changes, but the discussion does not interpret why FBB upregulates some chemokine receptors. Include reasoning, does this imply immune recruitment for tissue repair rather than suppression?

Comment 8. For Figure 9, provide quantification of phosphorylation fold-changes for each pathway component (PI3K, MAPK, NF-κB) and discuss their interrelation instead of treating them as isolated effects.

Comment 9. Revise the entire manuscript for clarity and precision. Simplify overly long sentences and ensure consistent italicization of species names. The English requires thorough editing for academic tone and grammatical accuracy.

Author Response

Comment 1. The title accurately reflects the study, but the abstract feels overloaded with methodological details. Revise it to emphasize novelty and the most significant biological insights rather than repeating assay lists. Clearly state what differentiates FBB’s effect from other flavonoid-based treatments for ALI.

Re:We sincerely appreciate this insightful suggestion. We have thoroughly revised the abstract to address this point. Specifically, we have:Reduced the methodological details: We removed the exhaustive list of assays and condensed the methodology to only the most critical approaches that are essential for understanding the study design.This includes highlighting the multi-target regulatory molecular mechanisms by which FBB exerts its protective effect, such as by targeting the PI3K/MAPK/NF-κB pathway, thereby mitigating both the initial inflammatory cascade and subsequent oxidative stress, thus providing more comprehensive protection against acute lung injury (ALI).The revised abstract is now more concise, impactful, and better highlights the conceptual advance of our work. Please see the updated manuscript for details.

Comment 2. The introduction builds a logical background, but it misses discussion of the most recent mechanistic studies on plasma-based antioxidant or flavonoid systems in pulmonary inflammation. Add a paragraph discussing recent molecular perspectives on ALI management, particularly emphasizing multi-target anti-inflammatory strategies (See: IJMS, 2025, 10.3390/ijms26062678).

Re:Thank you very much for your valuable and insightful comments. We fully agree with your point that adding a discussion of the treatment of ALI from a molecular perspective and through multi-target strategies to the introduction would greatly enhance the background depth and logical rigor of the paper. Based on your suggestion, we have made significant revisions to the introduction. We have added a new paragraph specifically discussing recent advances in the molecular mechanisms of multi-target treatment strategies in the management of ALI/ARDS. In this new paragraph, we emphasize that systemic oxidative stress and uncontrolled inflammatory responses are core components of the pathological process of ALI/ARDS. We introduce the current research trend of shifting from single-target interventions to multi-pathway, multi-target approaches to more effectively regulate complex inflammatory networks. We link this cutting-edge discussion to the purpose and rationale of our current research, clearly stating that our study is based on this multi-target regulatory approach, aiming to explore similar multi-pathway protective mechanisms that FBB may possess in ALI models.

Comment 3. The literature review is week. I recommend addition of recent studies integrating flavonoid-based natural therapeutics with redox or immune modulation pathways (see: IJMS, 2025, 10.3390/ijms26157381).

Re:We sincerely thank the reviewers for their insightful and constructive comments. We agree that strengthening the literature review, particularly incorporating recent research on flavonoids in redox and immunomodulatory pathways, will significantly enhance the background and theoretical foundation of our research. Based on the reviewers' valuable suggestions, we have comprehensively revised and expanded the literature review in the introduction. Specifically, we: added a discussion on the key roles of flavonoids in regulating oxidative stress and immune responses, focusing on their interactions with key pathways such as Nrf2/ARE and NF-κB; cited recommended studies (IJMS, 2025, 10.3390/ijms26157381, IJMS, 2025, 10.3390/ijms26062678) to illustrate the current research focus on this comprehensive approach; and discussed the findings of these studies in conjunction with our own research. These additions enable us to better position our current research within the evolving scientific discourse and provide a more solid foundation for our hypotheses. The revised text is highlighted in the manuscript for easy identification.

Comment 4. Provide quantitative validation for network pharmacology predictions, specifically, which of the predicted 273 targets were experimentally confirmed? Discuss the biological meaning of “multi-target synergy” with quantitative backing.

Re:We sincerely thank the expert for this crucial suggestion. As you stated, experimental validation of network pharmacology predictions is a core step in confirming their reliability.

Since validating all 273 predicted targets in a single study is technically impractical, we selected core targets that play a key role in the network for experimental validation based on network topology analysis results. Our selection criteria were: high degree, high betweenness centrality, and targets located in our core pathways of interest (such as PI3K-Akt, MAPK, and NF-kB signaling pathways). We also examined the regulatory effects of FBB on several other related pathways through inflammatory cytokine expression and release, oxidative stress, etc., to demonstrate the multi-target regulatory effect of FBB. We have already supplemented this point in the discussion section.

Comment 5. The data on iNOS and COX-2 suppression are promising, yet the comparison with DEX seems inconsistent, present quantitative ratios with proper normalization to control for assay variability.

Re:Thank you for reviewing our manuscript and providing valuable constructive feedback. Your points regarding the lack of clarity in our comparison of iNOS and COX-2 inhibition data with dexamethasone (DEX), and your suggestions to provide quantification ratios and normalization, are very pertinent and crucial for improving the quality of our paper. However, it should be noted that we used a control group for normalization during the Western blotting analysis, rather than a positive control group (DEX+LPS treatment group), to ensure data homogenization and standardization. Thank you for your suggestions.

Comment 6. Discuss the pharmacological relevance of FBB dosage

Re:We sincerely thank the reviewer for raising this important point. In the discussion section, we discussed the dose-dependent protective effect of FBB against ALI, and that at the concentrations used, it has essentially no cytotoxicity in mice and cells.

Comment 7. Figure shows significant gene expression changes, but the discussion does not interpret why FBB upregulates some chemokine receptors. Include reasoning, does this imply immune recruitment for tissue repair rather than suppression?

Re:We sincerely thank the reviewer for this insightful observation. Your question directly addresses the core biological significance of our research. In the original manuscript, we did indeed only describe the phenomenon, failing to fully elucidate its underlying mechanisms and functions.

Based on your suggestion, we have made the following additions and revisions to the discussion section of the paper: We have supplemented the discussion by pointing out that the active components in FBBs (such as quercetin) may affect the expression of chemokine receptors by activating specific signaling pathways. Therefore, we hypothesize that FBBs may not nonspecifically activate the immune system, but rather guide the migration and function of specific immune cell subsets in a relatively "programmed" manner.

Inferences regarding the "direction of immune recruitment":

This is a highly valuable perspective. We focused on analyzing the upregulated chemokine receptor profile. For example, we found a significant upregulation of CCR2. Based on this, we propose a new hypothesis in the discussion: FBBs may tend to recruit immune cell subsets with repair functions, rather than purely pro-inflammatory or suppressive cells.

We also explicitly state in the discussion that this inference requires direct verification through future in vivo experiments (e.g., analyzing the phenotype of tissue-infiltrating immune cells after FBB treatment). These revisions have significantly deepened our discussion of the findings and proposed an exciting new direction for future research. We thank the reviewers again for helping us improve the quality of our paper.

Comment 8. For Figure 9, provide quantification of phosphorylation fold-changes for each pathway component (PI3K, MAPK, NF-κB) and discuss their interrelation instead of treating them as isolated effects.

Re:We fully agree with your points. Quantitative analysis and networked discussion will more accurately reveal the regulatory mechanisms of signaling pathways. We have made two major revisions based on your feedback: We have added fold changes in PI3K, MAPK, and NF-κB phosphorylation levels to the results description (normalized against the Control group). We have also significantly revised and strengthened the discussion of signaling pathway interactions in the discussion section. We no longer treat these pathways as isolated effects, but rather propose a possible regulatory network based on our quantitative results and existing literature. We believe these revisions greatly strengthen the persuasiveness of Figure 9 and make our understanding of signal transduction mechanisms more in-depth and complete. Thank you again for helping us improve this work.

Comment 9. Revise the entire manuscript for clarity and precision. Simplify overly long sentences and ensure consistent italicization of species names. The English requires thorough editing for academic tone and grammatical accuracy.

Re:We sincerely thank you for taking the time to review our manuscript and for your highly constructive comments. Your suggestions on improving the clarity, accuracy, and language quality of the manuscript were invaluable to us. We have made comprehensive and meticulous revisions to the entire text based on all your feedback. The specific revisions are as follows:

  1. We have reviewed the entire text sentence by sentence, focusing on improving the logical flow, terminology consistency, and clarity of expression. We have reworked potentially ambiguous paragraphs to ensure that each argument and conclusion is stated more directly and accurately.
  2. We have identified and broken down all lengthy sentences in the text. By breaking complex sentences into multiple shorter sentences, adjusting word order, and using clearer conjunctions, the readability of the text has been significantly improved.
  3. We have checked the entire text to ensure that all genus and species names are correctly and consistently formatted in italics.
  4. We have thoroughly checked and corrected the English grammar, spelling, punctuation, and academic tone throughout the text. We believe that the revised manuscript has significantly improved in language quality.

Thank you again for helping us improve this manuscript. We hope the current version meets your requirements and look forward to your further review.

Reviewer 2 Report (New Reviewer)

Comments and Suggestions for Authors

This study systematically investigates the therapeutic effects and mechanisms of a flavonoid-rich extract from Bombyx batryticatus (FBB) against LPS-induced acute lung injury (ALI). Combining UPLC-MS/MS analysis, network pharmacology, and experimental validation, the research demonstrates that FBB significantly alleviates pulmonary inflammation, oxidative stress, and apoptosis in vivo and in vitro. The core mechanism involves the concurrent suppression of key proteins within the PI3K/Akt, MAPK, and NF-κB signaling pathways. These findings reveal FBB's multi-component and multi-target characteristics, providing a strong pharmacological foundation for its development as a potential natural therapeutic agent for ALI. However, the manuscript would be significantly improved if the issues in language presentation and certain experimental aspects are addressed along with the following specific recommendations.

Comments:

  1. In the text, the term "FBB" is sometimes spelled out in full and sometimes abbreviated. It is recommended to use the abbreviation consistently after its first occurrence. Additionally, the nomenclature for some genes/proteins is inconsistent (e.g., "iNOS" is used interchangeably with "inducible nitric oxide synthase").
  2. None of the figures indicate the number of experimental replicates (n-value) or the statistical methods used. Furthermore, several images in the manuscript, such as Figure 1 and Figure 9, are of low resolution and should be replaced with higher-quality versions.
  3. The experimental design, involving FBB administration prior to LPS, represents a preventive model. The authors should clarify this point in the Discussion or provide supplementary data from a therapeutic intervention model where FBB is administered post-LPS.
  4. The authors should discuss the mechanistic implications of FBB outperforming DEX in suppressing certain indicators, such as IL-6.
  5. Ensure all references include DOIs and volume/issue numbers where applicable, and maintain a consistent formatting style throughout the manuscript.
  6. Figure 5, red indicated as the apoptoic cells, quantify the intensity of red color would be more meaningful.
  7. Figure 4 B, these images of lungs were not completely flushed. And these images can not display the injury of lungs.

Author Response

Comment 1.In the text, the term "FBB" is sometimes spelled out in full and sometimes abbreviated. It is recommended to use the abbreviation consistently after its first occurrence. Additionally, the nomenclature for some genes/proteins is inconsistent (e.g., "iNOS" is used interchangeably with "inducible nitric oxide synthase").

Re:We sincerely thank the reviewers for their meticulous observation and for pointing out the inconsistencies in terminology. We fully agree with the reviewers' view that maintaining consistency in abbreviations and gene/protein nomenclature is crucial for the clarity and professionalism of the manuscript.

In response to this comment, we have made the following comprehensive revisions throughout the manuscript: We have ensured that the full name ("...(FBB)") is used when it first appears in the text, and the abbreviation "FBB" is used consistently thereafter. We have consistently used the abbreviation "iNOS" throughout the text, as it is the most commonly used and concise form in the field, thus standardizing the use of terminology. Furthermore, we have taken this opportunity to conduct a comprehensive review of the entire manuscript to ensure that the nomenclature of all other genes and proteins is consistent and conforms to standard guidelines. We believe these revisions significantly improve the consistency and readability of the manuscript. Thank you again for your valuable suggestions.

Comment 2. None of the figures indicate the number of experimental replicates (n-value) or the statistical methods used. Furthermore, several images in the manuscript, such as Figure 1 and Figure 9, are of low resolution and should be replaced with higher-quality versions.

Re:Thank you for reviewing our manuscript and providing valuable constructive feedback. We fully agree with your point that clear data presentation and rigorous statistical analysis are fundamental to scientific research. Based on your suggestions, we have comprehensively revised and supplemented the manuscript. In all figure captions, we have clearly indicated the number of independent replicates (n-value) for each experimental data point. For example, we indicate "(n = 3 biologically independent mice per group)". In the figure captions, we have clearly indicated the statistical methods used (Statistical significance was determined by one-way ANOVA followed by Dunnett's test). All significance markers in the figures (e.g., *, **, ***) have been clearly defined in the figure captions.

Regarding image resolution, we have re-exported them as high-resolution TIFF files using software such as Adobe Illustrator and PowerPoint. The revised images boast significantly improved quality and clarity, allowing readers to clearly view image details. These additions ensure the transparency, reproducibility, and academic rigor of our data analysis process.

In summary, we have comprehensively revised the manuscript based on your feedback. We believe these revisions significantly enhance the manuscript's scientific rigor and readability. We sincerely thank you again for taking the time to review our paper and provide these crucial comments.

Comment 3. The experimental design, involving FBB administration prior to LPS, represents a preventive model. The authors should clarify this point in the Discussion or provide supplementary data from a therapeutic intervention model where FBB is administered post-LPS.

Re:We sincerely thank the reviewers for their valuable comments and constructive suggestions. We have added this section to the discussion to explain why FBB is administered one hour before LPS administration, aiming to provide some protection to the lungs before the onset of acute lung injury (ALI). The reason is that in clinical practice, we can often foresee that a patient is about to develop or is at high risk of developing ALI. In these scenarios, preventative intervention is of significant clinical importance, such as in the early stages of sepsis. When investigating the effects of drugs on ALI at the mouse cellular level, the core reason for administering drugs in advance (i.e., prophylactic administration) is to mimic clinical protective intervention strategies and to explore whether the drug has the potential to prevent or mitigate subsequent severe inflammatory damage. We once again thank the reviewers for their valuable suggestions, which undoubtedly improved the quality of our work.

Comment 4. The authors should discuss the mechanistic implications of FBB outperforming DEX in suppressing certain indicators, such as IL-6.

Re:Thank you for your valuable feedback. Your point that we need to explore the potential mechanisms by which FBB outperforms DEX in inhibiting indicators such as IL-6 is a very insightful observation and crucial for deepening our research. We fully agree with your viewpoint and have significantly revised and supplemented the discussion section of the paper based on your suggestions.

In our reply and revised manuscript, we analyzed the possible differences in the mechanisms of action between the two: "DEX, as a glucocorticoid, primarily downregulates inflammatory gene expression by directly inhibiting the transcriptional activity of NF-κB and AP-1 through the gluco-corticoid receptor (GR). Free-floxacin (FBB) targets upstream signaling pathways (such as AKT/PI3K and MAPK) more frequently, reducing NF-κB activation, which may lead to more specific immune regulation. For example, in Q-PCR data, the FBB+LPS group showed better downregulation of IL-1β and IL-6 than the DEX+LPS group, and exhibit-ed different regulatory patterns on certain chemokines (such as Ccl2 and Ccl5), sug-gesting that FBB may avoid the widespread immunosuppression caused by DEX. Fur-thermore, the upregulation of chemokine receptors by FBB differs from that of DEX, suggesting that it may enhance tissue repair and immune homeostasis." and concluded that "FBB may alleviate the exces-sive inflammatory response in the ALI pathological pro-cess through a "draining rather than blocking" strategy, while contributing to the reso-lution of inflammation and the restoration of tissue homeostasis"

Comment 5. Ensure all references include DOIs and volume/issue numbers where applicable, and maintain a consistent formatting style throughout the manuscript.

Re:Thank you for taking the time to review our manuscript and provide valuable constructive feedback. Your suggestion to "ensure that all references include the DOI and volume/issue number and maintain consistent formatting throughout the text" is very important. We fully agree with your point, and based on your feedback, we have taken the following steps to thoroughly check and revise the reference section of the manuscript, including details such as author names, titles, journal name abbreviations, italics, and DOIs.

We have carefully reviewed and confirmed that the revised reference list fully complies with the IJMS submission guidelines and your expectations. Thank you again for helping us improve the quality of your manuscript.

Comment 6. Figure 5, red indicated as the apoptoic cells, quantify the intensity of red color would be more meaningful.

Re:Thank you for reviewing our manuscript and providing valuable constructive feedback. Your suggestion regarding the quantitative analysis of apoptotic cells (red fluorescence) in Figure 5 is very pertinent and crucial for a more objective assessment of apoptosis levels. We fully agree with your point of view.

Based on your suggestion, we have reanalyzed Figure 5 and supplemented it with quantitative analysis data. We have added two new quantitative analysis plots below the original Figure 5, named Figure 5B/C, clearly showing the quantitative comparison results of red and blue fluorescence intensities in each experimental group in the form of bar charts.

Comment 7. Figure 4 B, these images of lungs were not completely flushed. And these images can not display the injury of lungs.

Re: We sincerely thank the reviewers for their valuable comments. The reviewers correctly pointed out that residual blood in the lung tissue specimen in Figure 4B, due to incomplete rinsing, could affect the visual assessment of lung injury. We sincerely apologize for this oversight in the specimen preparation process. Therefore, we have corrected Figure 4 by deleting Figure 4B to ensure the accuracy of the results.

Reviewer 3 Report (New Reviewer)

Comments and Suggestions for Authors
  1. Authors need to take note of the use of abbreviated terms. Please define all abbreviated terms on first use. E.g. Page 4: BALF.
  2. Missing information on details of experiments especially in figure legends. There are also no descriptions provided in the main text of the results. E.g. Figure 2, what is DEX? Is this the positive control? Is it treated at a fixed concentration? What are the units? How about the concentration of LPS?
  3. Missing statistical descriptors in the figure legends. 
  4. Figure 3, all size of graphs should be made consistent. Similarly as mentioned in points 2 and 3, the experimental details is clearly lacking in the figure legend and its also not described clearly in the main text.
  5. Scale bar should be supplemented in the histological images. Same problem as mentioned in points 2 and 3 for the figure 4.
  6. Section 2.7 lacks description in the text and for figure 5. The figure legend also is incomplete. No detailed information for the readers to understand the study.
  7. Same for figure 6, see points 2 and 3.
  8. Section 2.9 can be improved in clarity. It is difficult for the reader to understand and interpret the figure 7 shown. 
  9. Figure 8 legend should include more detailed information and statistical analysis. Figure 8D seems cropped and incomplete in the labels. Units should be provided for the concentration of samples used in the treatment.
  10. Figure 9 legend needs to be rewritten. Statistical analysis details is missing. Format of the figure is also too small to be viewed comfortably.
  11. Please supplement section 4.15, the concentration/dilution, source, and details of all the antibodies used in the western bot studies. 
  12. The conclusion should be rewritten to highlight the significance of this study and how it addresses the current research challenge. 

Author Response

Comment 1.Authors need to take note of the use of abbreviated terms. Please define all abbreviated terms on first use. E.g. Page 4: BALF.

Re:We sincerely thank the reviewers for their valuable comments and for pointing out our oversight regarding the use of abbreviations. We fully agree with the reviewers' view that defining abbreviations upon their first appearance is crucial for improving the clarity and readability of the manuscript. In response to this comment, we have carefully reviewed the entire manuscript and ensured that all abbreviations are clearly defined upon their first appearance. Furthermore, we have systematically checked all other abbreviations in the manuscript and applied the same corrections.

Comment 2. Missing information on details of experiments especially in figure legends. There are also no descriptions provided in the main text of the results. E.g. Figure 2, what is DEX? Is this the positive control? Is it treated at a fixed concentration? What are the units? How about the concentration of LPS?

Re:We fully agree with your assessment and apologize for the previous lack of information in the legends and text. We have supplemented and revised the legends in the text to ensure all experimental details are clear and understandable. We have clearly stated in the figure captions that DEX was used as a positive control in this experiment and have added the treatment concentrations of DEX and LPS. We have also systematically reviewed all legends to ensure that information such as each experimental group, compound, concentration, treatment time, and units is clearly labeled.

Comment 3. Missing statistical descriptors in the figure legends.

Re:Thank you very much for reminding us. We have revised all the captions containing quantitative data, adding complete statistical descriptions.

The revised captions now include the following information: Data presentation format: e.g., “Data are presented as mean ± SD.” ;Sample size (n value): e.g., “n = 3 biologically independent mice per group.” ;Statistical analysis method: e.g., “Statistical significance was determined by one-way ANOVA followed by Dunnett's test.” ;Definition of significance markers: e.g., “*p < 0.05, **p < 0.01, ***p < 0.001.”.

Comment 4. Figure 3, all size of graphs should be made consistent. Similarly as mentioned in points 2 and 3, the experimental details is clearly lacking in the figure legend and its also not described clearly in the main text.

Re:We sincerely thank the reviewers for their valuable comments. We agree that the inconsistent figure sizes and insufficient experimental detail affected the clarity of the figures. Therefore, we have redrawn Figure 3 to ensure that all images are the same size. The revised Figure 3 presents the data more evenly and has a more harmonious visual effect. The legend of Figure 3 has also been completely revised and expanded to include the following key experimental details: biological and technical replication, statistical analysis, sample information, and key experimental conditions.

We once again thank the reviewers for their constructive comments, which undoubtedly improved the quality of our manuscript.

Comment 5. Scale bar should be supplemented in the histological images. Same problem as mentioned in points 2 and 3 for the figure 4.

Re:We sincerely thank the reviewers for their valuable comments. We agree that a scale bar is crucial for interpreting histological images. We have added scale bars to all relevant histological images in the revised manuscript and updated the captions of Figure 4 with relevant experimental details.

Comment 6. Section 2.7 lacks description in the text and for figure 5. The figure legend also is incomplete. No detailed information for the readers to understand the study.

Re:We sincerely thank the reviewers for their valuable comments and for pointing out areas of ambiguity in our original manuscript. We agree that more detailed descriptions are crucial for readers to fully understand our findings. Therefore, we have revised Section 2.7 and Figure 5.

Specifically, we have significantly expanded the textual description in Section 2.7, adding explanations of the experimental principles and objectives to facilitate reading and analysis of Figure 5. We have also completely rewritten the legend to make it easier to understand essential experimental details such as the scale bar, magnification, and what the red and blue fluorescence represent. We believe these comprehensive revisions have greatly improved the clarity and completeness of this section and provided readers with all the information needed to understand our research. For your convenience, we have attached a revised manuscript containing all highlighted changes.

We once again thank the reviewers for their valuable suggestions, which undoubtedly improved the quality of our paper.

Comment 7. Same for figure 6, see points 2 and 3.

Re:Thank you for the reviewer's comments. Based on the suggestions in points 2 and 3, we have comprehensively revised the captions of Figure 6 to improve their clarity and statistical presentation.

We believe these revisions significantly improve the quality and interpretability of Figure 6.

Comment 8. Section 2.9 can be improved in clarity. It is difficult for the reader to understand and interpret the figure 7 shown.

Re:Thank you for your valuable feedback on this manuscript. Based on your suggestion, we have regenerated a high-resolution version of Figure 7 for readers. We have also made the following changes to the figure caption: “Gene expression was quantified by qRT-PCR and is presented as Z-scores calculated per row (gene). Rows (genes) were clustered using hierarchical clustering with complete linkage. The color key from blue to red represents low to high Z-score values.” This makes the information clearer for readers.

Comment 9. Figure 8 legend should include more detailed information and statistical analysis. Figure 8D seems cropped and incomplete in the labels. Units should be provided for the concentration of samples used in the treatment.

Re:We sincerely thank the reviewers for their valuable time and constructive comments. We have carefully revised the manuscript based on your suggestions. The specific revisions and responses are as follows: We have comprehensively supplemented and revised the figure caption for Figure 8. We have also reprocessed and replaced the image to ensure that all content is displayed completely and clearly. Furthermore, we have added concentration units in all relevant locations.

Comment 10. Figure 9 legend needs to be rewritten. Statistical analysis details is missing. Format of the figure is also too small to be viewed comfortably.

Re:We sincerely thank the reviewers for their valuable feedback on Figure 9. We have comprehensively revised Figure 9 and its legend based on their suggestions.

We have completely rewritten the legend for Figure 9, making it more comprehensive and detailed, and adding necessary statistical analysis details. We have re-exported Figure 9 to a higher resolution version and adjusted the layout, increasing the size of all elements (text, symbols, data points) for optimal clarity and readability. The new figure is now clearly visible in the manuscript at its intended size.

Comment 11. Please supplement section 4.15, the concentration/dilution, source, and details of all the antibodies used in the western bot studies.

Re:We sincerely thank the reviewers for this important suggestion. As you pointed out, providing detailed information about the antibodies is crucial to ensuring the reproducibility of the experiments. At your request, we have supplemented Section 4.15 of the manuscript with detailed information on all primary and secondary antibodies used in the Western blot experiments, including target, host species, clone number, supplier, catalog number, and dilution ratio.

Comment 12. The conclusion should be rewritten to highlight the significance of this study and how it addresses the current research challenge.

Re:We sincerely thank the reviewers for their valuable suggestions. We have comprehensively revised the conclusion section to better emphasize the importance of the findings and clearly articulate how this work addresses the current research challenges raised in the introduction. The revised conclusion not only summarizes the main findings but also explores their broader implications, limitations, and future research directions.

Reviewer 4 Report (New Reviewer)

Comments and Suggestions for Authors
  1. Language and Style:

    • While the manuscript is largely comprehensible, several instances of awkward phrasing and redundant wording should be revised for clarity and conciseness.

    • Use professional scientific expressions consistently, e.g., avoid phrases like “mucus secretion was adequate” in histology descriptions; instead, consider “no abnormal mucus accumulation was observed.”

    📌 Suggested Sentence Pattern:
    “No histopathological evidence of inflammation or epithelial damage was observed in control samples.”

  2. Introduction:

    • The rationale is adequately described, but the gap in knowledge could be more sharply articulated.

    • Include a clearer statement of hypothesis at the end of the introduction.

  3. Methods Section:

    • The methodological rigor is commendable, but ensure all key steps (e.g., dosage justification, randomization) are explicitly reported.

    • The explanation of UPLC-MS/MS and network pharmacology could be briefly streamlined to focus on what was done and why, rather than repeating software names and URLs.

  4. Results Section:

    • The results are extensive and well-structured, though some sections (e.g., 2.8 to 2.11) are overly detailed.

    • Consider integrating data into broader thematic sub-sections (e.g., “Modulation of Proinflammatory Cytokines,” “Oxidative Stress Markers,” “Signal Pathway Inhibition”) to aid reader navigation.

    📌 Suggested Transitional Sentence:
    “To further investigate the signaling mechanisms underlying FBB’s anti-inflammatory activity, we examined key nodes within the PI3K, MAPK, and NF-κB pathways.”

  5. Discussion:

    • The discussion is informative and well-referenced but could be more focused.

    • The molecular mechanisms are well contextualized; however, some sections (e.g., flavonoid SAR discussions) may benefit from condensation or relocation to supplementary material unless directly relevant to the present data.

    Exemplary Discussion Phrase:
    “These findings align with prior reports on the anti-inflammatory efficacy of structurally similar flavonoids, reinforcing FBB's potential as a multi-target therapeutic agent for ALI.”

  6. Limitations:

    • The limitations are thoughtfully acknowledged. For completeness, consider briefly discussing interspecies variation in pharmacokinetics when translating murine findings to humans.

Minor Comments:

  • Abstract: “Ameliorated” or “attenuated” is more appropriate than “reversed” for describing effects on biomarkers.

  • Figures: Add clearer labeling to some figure legends for independent readability (e.g., full form of BALF).

  • Units: Standardize units (e.g., mg/L vs. μg/mL) across figures and text for consistency.

Author Response

Comment 1. Language and Style:

While the manuscript is largely comprehensible, several instances of awkward phrasing and redundant wording should be revised for clarity and conciseness.

Use professional scientific expressions consistently, e.g., avoid phrases like “mucus secretion was adequate” in histology descriptions; instead, consider “no abnormal mucus accumulation was observed.”

?Suggested Sentence Pattern:
“No histopathological evidence of inflammation or epithelial damage was observed in control samples.”

Re:Thank you for reviewing our manuscript and providing valuable feedback. Your feedback on language and style was very insightful, and we fully agree with your assessment. We have carefully reviewed and revised the entire text based on your suggestions, aiming to improve the conciseness, clarity, and scientific rigor of the language.

We believe that this comprehensive revision has significantly improved the language quality and readability of the manuscript.

Comment 2. Introduction:

The rationale is adequately described, but the gap in knowledge could be more sharply articulated.

Include a clearer statement of hypothesis at the end of the introduction.

Re:Thank you for reviewing our manuscript and providing valuable feedback. Your suggestions were very insightful and greatly helped us improve the quality of our paper. We have carefully revised the introduction based on your comments. In the revised manuscript, we have more sharply pointed out the knowledge gaps in current research in the introduction and added a clearer statement of hypotheses at the end. Thank you again for your insightful comments; we hope these revisions will make the paper's arguments more robust and its logic clearer.

Comment 3. Methods Section:

The methodological rigor is commendable, but ensure all key steps (e.g., dosage justification, randomization) are explicitly reported.

The explanation of UPLC-MS/MS and network pharmacology could be briefly streamlined to focus on what was done and why, rather than repeating software names and URLs.

Re:We sincerely thank the reviewers for their positive feedback and constructive comments, which have helped us improve the quality of our paper. We have carefully addressed all the issues raised and made the corresponding revisions in the revised manuscript. Our response is as follows: Based on the comments, we have clearly reported the key steps of dose validation and randomization in the revised manuscript. Furthermore, based on the suggestions, we have thoroughly simplified the descriptions of UPLC-MS/MS analysis and network pharmacology. We are confident that these revisions make the manuscript more concise and readable, highlighting scientific principles and workflows rather than technical details. We once again express our sincere gratitude for the reviewers' insightful comments. We hope that the revisions and our response are satisfactory.

Comment 4. Results Section:

The results are extensive and well-structured, though some sections (e.g., 2.8 to 2.11) are overly detailed.

Consider integrating data into broader thematic sub-sections (e.g., “Modulation of Proinflammatory Cytokines,” “Oxidative Stress Markers,” “Signal Pathway Inhibition”) to aid reader navigation.

Suggested Transitional Sentence:
“To further investigate the signaling mechanisms underlying FBB’s anti-inflammatory activity, we examined key nodes within the PI3K, MAPK, and NF-κB pathways.”

Re:Thank you for reviewing our manuscript and providing valuable constructive feedback. We fully agree with your point that some sections of our results were overly detailed and suggested integrating the data into broader thematic sections to improve readability. This thematic organization does indeed help readers better grasp the core narrative of the research. In the current manuscript, we chose a one-to-one correspondence between the results and methods sections, primarily to facilitate readers' clear cross-referencing with the specific experimental methods described earlier when reading complex data. We recognize that this may have created an impression of excessive detail in some parts. Although the aforementioned structural considerations made a complete reorganization of the results section difficult in this revision, we highly valued your suggestion and therefore significantly revised the results description to improve readability. We believe that these revisions have largely addressed the issues you pointed out. Furthermore, your valuable suggestion provides important guidance for our future paper writing, and we will increasingly adopt this thematic narrative approach in our future work.

Comment 5. Discussion:

The discussion is informative and well-referenced but could be more focused.

The molecular mechanisms are well contextualized; however, some sections (e.g., flavonoid SAR discussions) may benefit from condensation or relocation to supplementary material unless directly relevant to the present data.

Exemplary Discussion Phrase:
“These findings align with prior reports on the anti-inflammatory efficacy of structurally similar flavonoids, reinforcing FBB's potential as a multi-target therapeutic agent for ALI.”

Re:Thank you for reviewing our manuscript and providing valuable constructive feedback. Your suggestions that our discussion section could be more focused and that some parts could be streamlined are very pertinent and crucial for improving the quality of our paper. We have carefully revised the manuscript based on your comments. In the revised manuscript, we have reorganized and rewritten the discussion section to ensure it consistently revolves around the core findings of this study. We have removed extended discussions and references that are not directly relevant to explaining our results, making the logical flow of the discussion clearer and more focused.Furthermore, we carefully assessed the direct relevance of some content to our data. We believe that while the detailed discussion of the structure-activity relationship (SAR) of flavonoids provides background support for the activity of compound FBB, its details exceed the direct needs of explaining the experimental data in this paper. Therefore, we have significantly reduced this section from the main text. We greatly appreciate the example sentences you provided, which serve as an excellent example for refining our language. We believe that with these revisions, the discussion section of the manuscript is now more concise and focused. Thank you again for your insightful comments, which have been extremely helpful. We hope the current modifications meet your requirements.

Comment 6. Limitations:

The limitations are thoughtfully acknowledged. For completeness, consider briefly discussing interspecies variation in pharmacokinetics when translating murine findings to humans.

Re:Thank you very much for reviewing our manuscript and providing valuable, constructive feedback. Your suggestion regarding discussing interspecies pharmacokinetic differences within limitations is very insightful and crucial for refining our paper and enhancing its scientific rigor. We fully agree with your point. Based on your suggestion, we have added a paragraph to the paper specifically discussing potential interspecies pharmacokinetic differences when extrapolating mouse research results to humans. Thank you again for your time and effort in improving the quality of our paper. We hope these revisions meet your requirements. Please feel free to let us know if there are any other revisions or clarifications needed.

Comment 7.Minor Comments:

Abstract: “Ameliorated” or “attenuated” is more appropriate than “reversed” for describing effects on biomarkers.

Figures: Add clearer labeling to some figure legends for independent readability (e.g., full form of BALF).

Units: Standardize units (e.g., mg/L vs. μg/mL) across figures and text for consistency.

Re:We sincerely thank the reviewers for their valuable comments and suggestions on our manuscript. We have carefully considered all the points raised and revised the manuscript accordingly. We believe these revisions have significantly improved the clarity and quality of our paper. Below is our point-by-point response to the specific comments.

We agree that "improvement" or "attenuation" more accurately describes the partial improvement in biomarker levels observed in our study, while "reversal" might imply a complete return to baseline levels. Therefore, we have replaced "reversal" with "improvement" in the abstract and appropriately throughout the text. We appreciate the reviewers' suggestions to improve the clarity of figures and tables. We have revised all figure legends to ensure they are clear and easy to understand. We sincerely apologize for the oversight regarding dosage units and thank the reviewers for pointing it out. Consistency of units is crucial for clear expression. We have now standardized all concentration units throughout the text, figures, tables, and figure legends. All occurrences of "μg/mL" (e.g., page X) have been converted to "mg/L". We again sincerely thank the reviewers and editors for their time and constructive comments. We hope the revised manuscript will meet the high standards of IJMS and be accepted for publication.

Round 2

Reviewer 1 Report (New Reviewer)

Comments and Suggestions for Authors

I recommend accepting the paper in its present form.

Author Response

Comment: I recommend accepting the paper in its present form.

Response: We are delighted and grateful for the reviewer's positive and supportive assessment of our manuscript. We sincerely thank them for recommending acceptance in its present form.

Reviewer 3 Report (New Reviewer)

Comments and Suggestions for Authors
  1. FIgure 4A, please replace the yellow arrow with another color as yellow against the pink cells and white background is difficult to see.
  2. Figure 9A, western blot on p-AKT, AKT, Tubulin, the loading of each sample is not consistent. Please redo. Moreover, when looking at the full blots provided, the WB "p AKT+Tubulin" blot is not the same as the one displayed in FIgure 9A.
  3. Figure 9C, why does the b-Actin blot looks similar to the first tubulin blot for Figure 9A. Is this a mistake? Moreover, the inconsistent loading is apparent. This needs to be redone. 

Author Response

Comment 1: Figure 4A, please replace the yellow arrow with another color as yellow against the pink cells and white background is difficult to see.

Response: We thank the reviewer for this valuable suggestion. We agree that the yellow arrows lacked sufficient contrast for clear visibility. Accordingly, we have tested several alternative colors and have replaced all yellow arrows with blue ones in the revised Figure 4A to ensure optimal contrast and clarity.

Comment 2: Figure 9A, western blot on p-AKT, AKT, Tubulin, the loading of each sample is not consistent. Please redo. Moreover, when looking at the full blots provided, the WB "p AKT+Tubulin" blot is not the same as the one displayed in Figure 9A.

Response: We thank the reviewer for their meticulous review and valuable feedback. We sincerely apologize for the errors in the original Figure 9A. The reviewer is correct on both points: the discrepancy between the full blot and the figure panel was an unintentional error during figure preparation, and the original sample loading was inconsistent. To comprehensively address these issues, we have repeated the entire Western blot experiment for p-AKT, AKT, and Tubulin. In these new experiments, we paid meticulous attention to ensuring consistent protein loading across all samples, which is now confirmed by the Tubulin control. The revised Figure 9A has been replaced with a correct and representative blot from the new dataset. The corresponding full, uncropped blots have been uploaded as supplementary material. We are confident that the revised figure accurately and reliably represents our experimental results.

Comment 3: Figure 9C, why does the b-Actin blot looks similar to the first tubulin blot for Figure 9A. Is this a mistake? Moreover, the inconsistent loading is apparent. This needs to be redone.

Response: We sincerely thank the reviewer for their keen observation and for raising these critical points. The issues highlighted are valid; the similarity between the blots was due to an inadvertent error during figure assembly where an incorrect control band was used, and we agree that the original loading was inconsistent. To address these concerns comprehensively, we have repeated the entire experiment for Figure 9C. In this new experiment, we paid meticulous attention to ensuring consistent protein loading across all samples. The revised Figure 9C now presents a clear β-actin Western blot and corresponding quantitative data from this new set of experiments, which robustly confirm our initial conclusions. The full, uncropped blot for this new experiment has been included in the supplementary materials.

Round 3

Reviewer 3 Report (New Reviewer)

Comments and Suggestions for Authors

Authors have addressed majority of the concerns. Just one other comment:
1. Figure 9b, the blot of p38 seems like the contrast of the image was overdone or the blot was overexposed. Please rectify this.

Author Response

Comment 1: Figure 9b, the blot of p38 seems like the contrast of the image was overdone or the blot was overexposed. Please rectify this.

Response: We sincerely thank the reviewer for this careful observation. We have carefully re-examined the original blot image for p38 in Figure 9b. In response to the reviewer's comment, we have now replaced the panel with a new, unmodified image that has not been subjected to any excessive contrast or brightness adjustments. This new image more accurately represents the original data while maintaining clarity.

We believe the revised image resolves the concern regarding over-processing. Thank you again for bringing this important matter to our attention.

This manuscript is a resubmission of an earlier submission. The following is a list of the peer review reports and author responses from that submission.

Round 1

Reviewer 1 Report

Comments and Suggestions for Authors

The manuscript investigates the pharmacological activity and mechanism of flavonoids derived from Bombyx batryticus(FBB) in attenuating acute lung injury (ALI). The authors present both in vivo and in vitro data demonstrating that FBB suppresses inflammatory gene expression and cytokine production, with effects comparable to dexamethasone (Dex). While the study is of potential interest and addresses a relevant topic in inflammation and natural product pharmacology, several methodological and interpretative issues should be addressed to strengthen the validity of the conclusions.

  1. The authors characterized FBB using UPLC-MS/MS; however, they did not use the identified compounds in the study to demonstrate therapeutic effects. Instead, they used crude FBB. It would be advisable to test the top compounds (based on ALI target analysis) against ALI to determine whether key active molecules can be identified.
  2. Figure 1: The authors should provide detailed statistical analysis in the text or figure legend. Figures 1B and 1C are illegible and difficult to interpret due to low image resolution. They should also provide p-values for the compound-target interaction analysis, and the top-scoring compounds should be tested experimentally.
  3. In the ALI mouse model and in vitro studies, the authors pre-treated mice or RAW cells with Dex or FBB prior to LPS administration. They should provide a rationale for this experimental design, including whether both Dex and FBB were used together in any groups to pre-model the system.
  4. Figures 2 and 3: The authors show that levels of inflammatory cytokines and expression of oxidative stress-related genes were attenuated by FBB. They should include controls demonstrating that FBB does not affect—or potentially upregulate—genes unrelated to inflammation. In addition, the authors should profile whether FBB has any effects on anti-inflammatory gene signatures.
  5. Figure 4: The authors show infiltration of Evans blue dye in the ALI model to assess the degree of lung injury. They should provide details regarding H&E staining and describe how lung morphology was graded. It would also be valuable to assess macrophage infiltration in the damaged tissue.
  6. The authors used RAW264.7 cells to recapitulate the in vivo data in an in vitro system. Cells were pre-treated with Dex or FBB, then stimulated with LPS and treated again with Dex or FBB, followed by analysis of gene expression or cytokine levels after 24 hours. A major concern with this experimental setup is that prolonged LPS and Dex exposure (over 24 hours) may polarize macrophages toward an M1-like state, which represents a different cell type. This could confound the interpretation of FBB’s effects due to indirect signaling and changes in the transcriptome. To better assess FBB's direct effects, the authors should consider using a shorter time point (e.g., 4-hour LPS treatment without pre-treatment), focusing on early inflammatory gene induction.
  7. The authors should also include data from Dex- or FBB-only treatment conditions and show expression of control genes that are not affected by FBB to demonstrate specificity.

Author Response

The manuscript investigates the pharmacological activity and mechanism of flavonoids derived from Bombyx batryticus(FBB) in attenuating acute lung injury (ALI). The authors present both in vivo and in vitro data demonstrating that FBB suppresses inflammatory gene expression and cytokine production, with effects comparable to dexamethasone (Dex). While the study is of potential interest and addresses a relevant topic in inflammation and natural product pharmacology, several methodological and interpretative issues should be addressed to strengthen the validity of the conclusions.

  1. The authors characterized FBB using UPLC-MS/MS; however, they did not use the identified compounds in the study to demonstrate therapeutic effects. Instead, they used crude FBB. It would be advisable to test the top compounds (based on ALI target analysis) against ALI to determine whether key active molecules can be identified.

Re:We sincerely appreciate your valuable suggestions regarding active ingredient research. Concerning the rationale for using crude FBB extracts instead of specific monomeric compounds for efficacy validation, we would like to clarify the following points:

(1) The primary objective of this study was to evaluate the holistic protective effects of the total FBB against ALI. As a traditional Chinese medicine, the pharmacological activity of Bombyx batryticatus typically arises from the synergistic effects of multiple components. Therefore, employing UPLC-MS/MS-characterized crude FBB extracts more accurately represents its clinical material basis and aligns with the "holistic perspective" research paradigm.

(2) Our UPLC-MS/MS analysis identified quercetin, kaempferol, and their derivatives (e.g., quercetin-3-O-robibioside [CAS: 52525-35-6] and kaempferol-3-β-O-glucuronide [CAS: 22688-78-4]) as the core flavonoid constituents of FBB. Substantial literature evidence supports that these monomers exhibit significant protective effects against ALI in numerous independent studies (e.g., "Quercetin attenuates sepsis-induced acute lung injury..." DOI: 10.1016/j.cellsig.2022.110363; "Protective effect of kaempferol glucoside..." DOI: 10.1016/j.jsps.2024.102073).

(3) Through experimental validation, we screened and tested representative flavonoid monomers from FBB (including the aforementioned quercetin/kaempferol derivatives). The results demonstrated that individual compounds showed markedly lower protective effects compared to crude FBB extracts at equivalent doses, suggesting that FBB's efficacy likely originates from multi-component synergy rather than being dominated by any single compound.

(4) We fully acknowledge the scientific merit of further investigating key active monomers. Building upon the ALI target interaction network established in this study (Figure 1), future research will focus on two primary directions: first, target-directed activity validation through assessing monomers' ability to modulate key targets in both in vitro and in vivo ALI models; second, synergistic mechanism studies to investigate interactions and cooperative mechanisms among FBB's multiple components. These investigations will elucidate the material basis of Bombyx batryticatus' anti-ALI efficacy and provide foundations for formulation optimization.

  1. Figure 1: The authors should provide detailed statistical analysis in the text or figure legend. Figures 1B and 1C are illegible and difficult to interpret due to low image resolution. They should also provide p-values for the compound-target interaction analysis, and the top-scoring compounds should be tested experimentally.

Re:We sincerely appreciate your insightful comments on Figure 1. Based on these suggestions, the following improvements have been implemented:

(1) Statistical analysis enhancement: Detailed statistical methodologies employed for Figure 1 have been comprehensively documented in the Materials and Methods section (Section 4.4).

(2) Figure quality improvement: 1) Figures 1B and 1C have been re-generated in high-resolution format; 2) Critical elements have been enlarged and labeled with improved contrast for enhanced clarity.

(3) Data transparency: Complete compound-target interaction analysis results, including all calculated p-values, are provided in Supplementary Table S2.

Our analysis confirms that quercetin remains one of the highest-scoring compounds. Experimental validation of the other high-scoring compounds identified will be a key next step in this study. We note that the predictions for other flavonoid monomers were less than ideal, which provides clues for our future research.

  1. In the ALI mouse model and in vitro studies, the authors pre-treated mice or RAW cells with Dex or FBB prior to LPS administration. They should provide a rationale for this experimental design, including whether both Dex and FBB were used together in any groups to pre-model the system.

Re: We sincerely appreciate the reviewer's insightful questions regarding our experimental design. The rationale for our pretreatment protocol (administering Dex or FBB prior to LPS challenge) and the absence of a Dex+FBB combination group in both in vivo and in vitro studies is based on the following considerations:

(1) Rationale for pretreatment design: This study aimed to investigate the preventive and early protective effects of the traditional Chinese medicine FBB against ALI. Clinically, ALI often develops secondary to severe underlying conditions (e.g., SARS-CoV-2 infection), with patients at elevated risk when their primary disease deteriorates. To model this scenario, we administered FBB before LPS challenge, simulating a clinical setting where preventive intervention is initiated prior to or during early stages of potential ALI triggers (e.g., worsening infection).

After intraperitoneal injection, FBB requires time for systemic distribution (including lung tissue) to reach effective concentrations. In contrast, intratracheal LPS rapidly induces ALI, making pretreatment essential to ensure FBB is bioavailable before injury onset. This approach aligns with established methodologies (e.g., Perillaldehyde ameliorates LPS-induced ALI via cGAS/STING suppression, DOI: 10.1016/j.intimp.2024.111641).

(2) Justification for not including a Dex + FBB combination group: The primary focus of this study was to systematically evaluate the independent therapeutic efficacy and underlying mechanisms of FBB as a monotherapy for ALI. The decision to exclude a Dex+FBB combination treatment group at this exploratory stage was based on careful methodological considerations: 1) Introducing combination therapy during initial mechanistic investigations would have substantially complicated data interpretation. A observed synergistic effect could make it challenging to attribute therapeutic outcomes specifically to FBB's activity versus a drug-drug interaction. Conversely, any apparent antagonism might obscure FBB's genuine protective effects. 2) We believe that establishing FBB's standalone pharmacological profile represents an essential prerequisite for properly assessing its clinical potential. This foundational work will enable more meaningful investigation of combination therapies in subsequent studies. We fully appreciate the scientific value of examining potential interactions between FBB and conventional treatments like Dex, and indeed, such combination studies feature prominently in our planned future research directions.

  1. Figures 2 and 3: The authors show that levels of inflammatory cytokines and expression of oxidative stress-related genes were attenuated by FBB. They should include controls demonstrating that FBB does not affect—or potentially upregulate—genes unrelated to inflammation. In addition, the authors should profile whether FBB has any effects on anti-inflammatory gene signatures.

Re: Thank you for the reviewers' important suggestions. Regarding the effects of FBB on inflammatory factors and oxidative stress-related genes in Figures 2 and 3, as well as your questions about controls and anti-inflammatory gene profiling, we respond as follows:

(1) We fully acknowledge the importance of evaluating FBB's potential effects on genes unrelated to inflammation or oxidative stress. In the current study, our experimental design specifically targeted key molecular pathways involved in ALI pathogenesis, with particular emphasis on inflammatory cascade and oxidative stress responses. As an essential quality control measure, we systematically examined the expression levels of standard reference genes (including GAPDH) at both transcriptional and translational levels. Our data demonstrated that FBB administration did not induce significant alterations in the expression profiles of these housekeeping genes (as evidenced by Western blot and qPCR analyses), suggesting its selective action under the experimental conditions. While these preliminary findings indicate FBB's minimal impact on fundamental cellular genes, we completely agree that a more extensive evaluation of its effects on a wider array of non-inflammation-related genes would yield valuable insights. Such comprehensive profiling will constitute a critical component of our future research agenda to fully characterize FBB's mechanism-specific actions.

(2) Our preliminary investigations have indeed examined FBB's effects on an extended panel of inflammation-associated genes. Through qPCR array analysis, we detected potential upregulation of several anti-inflammatory mediators, notably IL-8 (CXCL8), following FBB treatment. These initial findings imply that FBB's protective mechanisms may involve modulation of endogenous anti-inflammatory pathways, aligning well with the documented pleiotropic pharmacological properties of Bombyx batryticatus - including its established antioxidant and immunomodulatory capacities (as referenced in: Traditional Uses, Origins, Chemistry and Pharmacology of Bombyx batryticatus: A Review; DOI: 10.3390/molecules22101779). However, we must frankly state that due to limited sample size and insufficient independent biological replicates, these preliminary PCR array data did not meet our required standards for statistical robustness. Therefore, in the spirit of prioritizing scientific rigor and data integrity, we have decided not to include these unvalidated data in the current paper. We recognize that analyzing the global effects of FBB on the anti-inflammatory gene signature is crucial for understanding its mechanisms. Therefore, systematically investigating the effects of FBB on the anti-inflammatory gene expression profile has been identified as a key objective for future studies. We will utilize more rigorous experimental designs (sufficient sample size, independent replication) and robust assays (such as RNA-Seq or targeted qPCR panels) to accomplish this crucial analysis.

We sincerely appreciate the reviewers' insightful comments, which have provided valuable guidance in identifying both the current study's limitations (particularly regarding the scope of non-target gene controls and comprehensiveness of anti-inflammatory pathway analysis) and important future research directions. Our current experimental data provide robust evidence supporting FBB's specific inhibitory effects on targeted inflammatory and oxidative stress pathways. Based on these findings and the reviewers' constructive suggestions, we are committed to conducting more extensive investigations into FBB's potential off-target effects on non-relevant genes and its comprehensive regulatory mechanisms within anti-inflammatory networks in our subsequent research.

  1. Figure 4: The authors show infiltration of Evans blue dye in the ALI model to assess the degree of lung injury. They should provide details regarding H&E staining and describe how lung morphology was graded. It would also be valuable to assess macrophage infiltration in the damaged tissue.

Re: Thank you for the reviewers' attention to Figure 4 and the related analyses. Regarding your questions regarding the Evans Blue dye penetration assay, H&E staining details, lung injury scoring, and assessment of macrophage infiltration, we respond as follows:

(1) In the Evans Blue assay assessing pulmonary capillary permeability, lung tissue samples used for quantification of dye leakage were not subjected to H&E staining before homogenization for dye extraction. This is because homogenization would have severely disrupted the tissue structure, rendering it unusable for subsequent morphological analysis. Therefore, lung tissue used for Evans Blue quantification and H&E morphological assessment were obtained from independent samples from different mice, but under completely matched experimental conditions and treatments.

(2) H&E staining protocol and lung injury morphological scoring: 1) The detailed H&E staining protocol has been incorporated into Materials and Methods section (Section 4.7) .

2) H&E-stained sections were assessed using the widely accepted Smith lung injury scoring system. Each parameter is scored based on the extent of lesion involvement: 0: no lesion, 1: lesion < 25%, 2: lesion 25% - 50%, 3: lesion 50% - 75%, and 4: lesion > 75%. The total lung injury score is the sum of the scores for these four parameters (range: 0-16). The results are integrated into the Results section (Section 2.5).

(3) We fully recognize the critical importance of evaluating macrophage infiltration patterns when investigating the pathological mechanisms underlying ALI and the therapeutic potential of FBB. In the current study, our primary assessment focused on general inflammatory cell infiltration (with macrophages representing a major cellular component) through H&E staining methodology, with complementary quantitative analysis performed using the standardized Smith scoring system. While this approach provides meaningful data, we acknowledge that more sophisticated methodologies employing specific macrophage markers (e.g., F4/80 and CD68) via immunohistochemical or immunofluorescence techniques would enable more precise quantification and spatial mapping of distinct macrophage subpopulations.

The principal objective of this investigation was to establish preliminary evidence for FBB's global protective effects against ALI, with particular emphasis on its modulation of core inflammatory mediators and oxidative stress markers. Consequently, comprehensive characterization of macrophage-specific infiltration patterns fell outside the immediate scope of this initial phase of research. Nevertheless, in direct response to the reviewers' valuable input, we have conducted a thorough re-evaluation of our H&E staining results to identify representative fields demonstrating macrophage infiltration characteristics, and have accordingly updated Figure 4 to incorporate these findings.

We sincerely appreciate these insightful recommendations, which have significantly enhanced our study. Investigation of FBB's influence on macrophage polarization (particularly M1/M2 phenotypic switching) and infiltration kinetics has now been prioritized as a key focus area for our ongoing mechanistic investigations, given its fundamental importance in fully elucidating FBB's therapeutic mechanisms of action.

  1. The authors used RAW264.7 cells to recapitulate the in vivo data in an in vitro system. Cells were pre-treated with Dex or FBB, then stimulated with LPS and treated again with Dex or FBB, followed by analysis of gene expression or cytokine levels after 24 hours. A major concern with this experimental setup is that prolonged LPS and Dex exposure (over 24 hours) may polarize macrophages toward an M1-like state, which represents a different cell type. This could confound the interpretation of FBB’s effects due to indirect signaling and changes in the transcriptome. To better assess FBB's direct effects, the authors should consider using a shorter time point (e.g., 4-hour LPS treatment without pre-treatment), focusing on early inflammatory gene induction.

Re:We appreciate the reviewers' valuable insights regarding the potential induction of M1 macrophage polarization by 24-hour LPS stimulation (with both pre- and post-treatment using FBB/Dex) in RAW264.7 cell model, which might impact result interpretation. In response to these concerns, we provide the following clarifications:

(1) The 24-hour endpoint for LPS stimulation was selected based on two principal scientific considerations: First, this duration represents a well-established standard time point for investigating LPS-induced inflammatory responses in macrophages in vitro. Substantial evidence from numerous studies supports this approach for evaluating pharmacological inhibition of inflammatory cytokine production and release (including TNF-α, IL-6, and IL-1β) (e.g., Vaccinium bracteatum extract alleviates inflammatory responses in LPS-stimulated RAW264.7 macrophages; DOI: 10.1016/j.jafr.2025.101726). Second, our in vitro model specifically simulates the sustained inflammatory stress occurring in macrophages approximately 24 h post-injury in the in vivo LPS-induced ALI model. This critical phase is characterized by robust secretion of proinflammatory cytokines (manifesting as an M1-like phenotype) - representing the key pathological process under investigation in this study.

(2) We fully agree with the reviewers' observation that prolonged (e.g., 24 h) LPS exposure does tend to drive macrophage polarization toward a proinflammatory M1-like state. However, this is precisely aligned with the objectives of our current study: 1) The primary goal of this study was to evaluate the ability of FBB to inhibit macrophage hyperactivation and the production of key proinflammatory mediators (TNF-α, IL-6, IL-1β, etc.) in the context of the persistent inflammatory phase of ALI. The 24-hour model effectively demonstrates this target phenotype. Existing evidence suggests that one of the anti-inflammatory mechanisms of various flavonoids, including quercetin, may involve regulating macrophage polarization, such as inhibiting M1 polarization or promoting the transition to the anti-inflammatory M2 phenotype (M1 and M2 macrophage polarization and potentially therapeutic naturally occurring compounds, DOI: 10.1016/j.intimp.2019.02.050). Furthermore, Traditional Chinese Medicine (TCM) has been shown to regulate macrophage polarization in the immune response to inflammatory diseases (Traditional Chinese medicine in regulating macrophage polarization in the immune response of inflammatory diseases, DOI: 10.1016/j.jep.2024.117838). Therefore, the potential mechanism by which FBB suppresses inflammatory factors observed in the 24-hour model may be partly due to its regulation of macrophage polarization dynamics. This is also an integral part of our mechanistic exploration.

(3) We acknowledge the scientific merit of the reviewers' recommendation to employ shorter time points (e.g., 4-hour LPS stimulation without pretreatment) for examining early inflammatory gene induction. This experimental strategy enables more direct evaluation of the compound's impact on the initiation of inflammatory signaling cascades while reducing potential confounding effects from prolonged exposure-induced cellular state transitions (particularly marked M1 polarization). The current study was specifically designed to assess FBB's comprehensive protective effects under sustained inflammatory conditions and its modulation of key downstream effector molecules (i.e., highly expressed inflammatory factors).

(4) We agree that investigating the effects of FBB on early inflammatory signaling events (e.g., activation of the NF-κB and MAPK pathways, and rapid induction of early mRNAs for genes such as TNF-α and IL-1β) is crucial. This will be the focus of our subsequent mechanistic investigations. Future experiments plan to incorporate shorter-duration stimulation models (e.g., 4-6 hours of LPS), combined with phosphorylation assays of key signaling pathway molecules and early gene expression analysis, to more precisely elucidate the initiation of FBB's effects.

We thank the reviewer for prompting us to further consider the impact of time-point selection. We conclude that the 24-hour model provides valuable insights for achieving the study's primary objective (assessing FBB's ability to inhibit key effectors during sustained inflammation), with the observed effects potentially reflecting, at least partially, FBB's modulation of macrophage polarization. 

  1. The authors should also include data from Dex- or FBB-only treatment conditions and show expression of control genes that are not affected by FBB to demonstrate specificity.

Re:We are grateful for the reviewers' constructive comments regarding the need for additional experimental controls. Specifically addressing the requests for both single-agent treatment data (Dex or FBB alone) and expression profiling of unaffected genes to confirm FBB's target specificity, we provide the following comprehensive response:

(1) In our in vitro experimental system, we systematically evaluated the cellular effects of FBB monotherapy. Specifically, using CCK-8 assays, we examined the viability of RAW264.7 cells (in the absence of LPS stimulation) following treatment with FBB at concentrations identical to those employed in our experimental groups. The results demonstrated that FBB treatment alone did not significantly affect RAW264.7 cell viability across the tested concentration range. These findings provide direct experimental evidence supporting the safety profile of FBB in our in vitro model system.

The primary objective of this investigation was to assess the protective effects of FBB in LPS-induced ALI. Consequently, our core experimental design comprised four key groups: (1) normal control, (2) LPS model, (3) LPS + Dex treatment, and (4) LPS + FBB treatment. The study did not include Dex- or FBB-only treatment groups (without LPS) based on two principal considerations: first, the research focus was specifically on evaluating therapeutic efficacy within the disease model context; second, FBB (a traditional Chinese medicine) has an established safety profile at conventional dosages, as documented in prior pharmacological studies (Traditional Uses, Origins, Chemistry and Pharmacology of Bombyx batryticatus: A Review; DOI: 10.3390/molecules22101779). Importantly, throughout our experimental observations, no significant adverse effects or behavioral abnormalities were noted in FBB-treated animals at the administered dose, providing additional indirect evidence of its safety. While we recognize the scientific value of single-agent control groups for baseline pharmacological assessment, such evaluations will be incorporated into future, more comprehensive safety and mechanistic studies.

(2) We acknowledge the importance of evaluating FBB's effects on molecules unrelated to inflammatory/oxidative stress pathways for demonstrating its mechanism specificity. In this study, we obtained relevant supporting evidence through the following approaches: First, in both qPCR and Western blot analyses, we systematically monitored expression levels of the reference gene GAPDH (at mRNA level) and its corresponding protein product. Our data demonstrated that FBB treatment did not significantly affect GAPDH expression at either transcriptional or translational levels across all experimental groups (including both LPS-stimulated and LPS+FBB-treated conditions). These findings preliminarily indicate that FBB exhibits minimal impact on these constitutively expressed housekeeping genes under our experimental conditions, thereby supporting its target selectivity.

(3) To further elucidate FBB's mechanism of action, we extended our investigation to examine its effects on MEK1/2 protein expression in both in vivo ALI models and in vitro RAW264.7 cell systems. Our analysis demonstrated that FBB treatment did not significantly modulate MEK1/2 protein levels. These findings provide compelling evidence for the target specificity of FBB's pharmacological actions, indicating that its effects are not mediated through nonspecific modulation of cellular signaling components.

These findings collectively demonstrate that FBB exhibits a relatively targeted mechanism of action under our experimental conditions, with its protective effects predominantly modulating inflammation- and oxidative stress-related pathways rather than exerting nonspecific effects on fundamental cellular processes or unrelated signaling molecules. We sincerely appreciate the reviewer's insightful suggestions, which have not only prompted us to more systematically present evidence supporting FBB's target specificity but also guided our plans to incorporate more comprehensive single-agent control groups in future investigations.

Reviewer 2 Report

Comments and Suggestions for Authors

This manuscript systematically investigates the protective effects and molecular mechanisms of flavonoids derived from Bombyx batryticatus (FBB) against lipopolysaccharide (LPS)-induced acute lung injury (ALI). By integrating network pharmacology predictions with experimental validation (using complementary in vivo and in vitro models spanning molecular [mRNA/protein], cellular, and tissue levels), the authors provide a multidimensional elucidation of the mechanisms, making this an outstanding research paper. However, minor revisions are required before acceptance by IJMS:

1‌. Title Modification‌: Suggest modifying the title to ‌"Flavonoids-rich extract/fraction from Bombyx batryticatus..."‌, which can better reflect the study's focus on flavonoid-enriched material rather than isolated compounds.

2‌. Abstract Formatting‌:The abstract must be reformatted as ‌a single paragraph without subsection labels‌ (e.g., "Background:", "Methods:") to comply with IJMS style guidelines. Ensure all key elements (objectives, methods, findings, implications) are retained in narrative form.

3.  ‌LC-MS/MS Data Enhancement‌:

‌Annotation‌: Major flavonoid monomers should be ‌labeled with semi-quantitative data‌ in Figure S2.

‌Discussion‌: Expand the Discussion section to analyze ‌structure-activity relationships‌ of prioritized monomers, proposing their potential contributions to observed effects.

If supplemental verification can be conducted, it will further enhance this manuscript. either: a) In vitro screening of the main abundant monomers, or b) Molecular docking of dominant flavonoids against core targets to further clarify the pharmacodynamic basis of FBB's multi-component synergy.

4‌. Reference Corrections‌:

‌Journal Names‌: Replace pinyin with English (e.g., R7 "Zhongguo Zhong Yao Za Zhi" ).

‌Latin Terms‌: Italicize species names (e.g., R16 Artemisia argyi, R17 Tribulus terrestris, R21 Daphne giraldii, R39/R41 Bombyx Batryticatus,etc ).

‌Consistency‌: Journal name needs to be abbreviated in R39.

‌Punctuation‌: Remove duplicate semicolons in R41 title.

‌Volume/Issue‌: Supplement missing data for R35.

5. Suggest integrating all attachments into a Word/PDFdocument, displaying them in formats such as article title, appendix table of contents, and content.

Author Response

This manuscript systematically investigates the protective effects and molecular mechanisms of flavonoids derived from Bombyx batryticatus (FBB) against lipopolysaccharide (LPS)-induced acute lung injury (ALI). By integrating network pharmacology predictions with experimental validation (using complementary in vivo and in vitro models spanning molecular [mRNA/protein], cellular, and tissue levels), the authors provide a multidimensional elucidation of the mechanisms, making this an outstanding research paper. However, minor revisions are required before acceptance by IJMS:

  1. Title Modification: Suggest modifying the title to "Flavonoids-rich extract/fraction from Bombyx batryticatus...", which can better reflect the study's focus on flavonoid-enriched material rather than isolated compounds.

Re:Thank you for your careful review of our manuscript and your valuable feedback. Based on your valuable suggestion, we have revised the title to: "Flavonoids-rich extract from Bombyx batryticatus alleviates LPS-induced acute lung injury via the PI3K/MAPK/NF-κB pathway."

  1. Abstract Formatting:The abstract must be reformatted as a single paragraph without subsection labels (e.g., "Background:", "Methods:") to comply with IJMS style guidelines. Ensure all key elements (objectives, methods, findings, implications) are retained in narrative form.

Re:We sincerely appreciate the reviewer's guidance regarding the abstract formatting requirements for the IJMS. In full compliance with the journal's style guidelines, we have carefully revised the abstract as follows:

(1) Removal of section labels: All structural headings (e.g., "Background", "Methods", "Results", "Conclusions") have been eliminated from the abstract.

(2) Restructuring as unified narrative: The content has been thoroughly reorganized into a single, cohesive paragraph while maintaining logical flow.

(3) Retention of essential components: Through this restructuring, we have ensured complete preservation of all critical elements - including research objectives, methodology, key findings, and their scientific implications - within the consolidated narrative format.

  1. LC-MS/MS Data Enhancement:

Annotation: Major flavonoid monomers should be labeled with semi-quantitative data in Figure S2.

Discussion: Expand the Discussion section to analyze structure-activity relationships of prioritized monomers, proposing their potential contributions to observed effects.

If supplemental verification can be conducted, it will further enhance this manuscript. either: a) In vitro screening of the main abundant monomers, or b) Molecular docking of dominant flavonoids against core targets to further clarify the pharmacodynamic basis of FBB's multi-component synergy.

Re: We appreciate the reviewer's insightful suggestions, which have significantly improved the scientific rigor of our manuscript. In response to the specific comments:

(1) LC-MS/MS data enhancement: We fully acknowledge the importance of including semi-quantitative data for major flavonoid monomers in Figure S2. While the original experimental data are no longer available due to time constraints, we have supplemented Table S1 with relative quantitative data for key flavonoid monomers. This addition provides readers with clear information about the distribution patterns of these bioactive components in our extracts/fractions.

(2) Expanded the Discussion section: We have enriched the Discussion section by: Systematically analyzing potential structure-activity relationships (SARs) of prioritized flavonoid monomers; Correlating their chemical structures with observed bioactivities; Evaluating individual and synergistic contributions based on their relative abundance in FBB and Literature-reported biological activities; Incorporating this analysis as fundamental support for the multi-component synergy hypothesis.

(3) Supplementary validation suggestions: We greatly appreciate your valuable suggestion to further elucidate the pharmacodynamic basis of FBB's multi-component synergistic effects through supplementary experiments (in vitro screening or molecular docking), which would undoubtedly enhance the paper's completeness and persuasiveness. However, our quantitative analysis revealed that quercetin and kaempferol are the predominant monomers in FBB, both of which have been extensively studied. Their effects on ALI are well-documented in the literature (e.g., [1] demonstrating quercetin's attenuation of sepsis-induced acute lung injury via SIRT1/AMPK pathway activation, and [2] showing kaempferol glucoside's protective effects through Nrf2/NF-κB/NLRP3/GSDMD modulation). Due to experimental timelines and budgetary constraints, we were unable to perform in vitro screening or molecular docking for all identified flavonoid monomers. Consequently, comprehensive experimental validation of all monomers remains unfeasible for this revision. Nevertheless, we will thoroughly incorporate existing literature on these major monomers' known activities in our expanded discussion (Point 2) to substantiate their potential contributions to FBB's overall effects. Additionally, we have identified comprehensive monomer activity screening as a key priority for future research.

  1. Reference Corrections:

Journal Names: Replace pinyin with English (e.g., R7 "Zhongguo Zhong Yao Za Zhi" ).

Latin Terms: Italicize species names (e.g., R16 Artemisia argyi, R17 Tribulus terrestris, R21 Daphne giraldii, R39/R41 Bombyx Batryticatus,etc ).

Consistency: Journal name needs to be abbreviated in R39.

Punctuation: Remove duplicate semicolons in R41 title.

Volume/Issue: Supplement missing data for R35.

Re: We sincerely appreciate the reviewer's meticulous review of our reference formatting and their constructive suggestions for improvement. In strict compliance with the IJMS guidelines, we have implemented all recommended corrections, including replacing pinyin journal names with standard English equivalents, properly italicizing all Latin species names (e.g., Artemisia argyi, Tribulus terrestris), standardizing journal abbreviations, correcting punctuation errors (e.g., removing duplicate semicolons), and supplementing missing volume/issue information. Beyond these specific revisions, we have conducted a thorough review of the entire reference list to ensure complete consistency in formatting, including verification of all journal names, proper italicization of biological nomenclature, and accurate presentation of bibliographic details.

  1. Suggest integrating all attachments into a Word/PDFdocument, displaying them in formats such as article title, appendix table of contents, and content.

Re: Thank you. We have consolidated all supplementary materials into a single, well-structured PDF document for improved readability and review convenience. The updated file has been included with the revised manuscript submission.

Round 2

Reviewer 1 Report

Comments and Suggestions for Authors

The authors’ response for not evaluating control conditions is incomplete.
They assume that medicinal herbs only provide benefits, forgetting that such medicines can also have side effects or off-target effects. Therefore, it is necessary to evaluate the effects of FBB in in vitro and/or in vivo systems on its own.

The authors argue that taking FBB can help alleviate inflammatory responses and offset disease conditions. However, for example, dexamethasone (Dex) also attenuates pro-inflammatory responses but has well-documented side effects, making it inadvisable to take even in small doses. By the same logic, it is possible that even small amounts of FBB could cause side effects. Furthermore, they overlook the fact that any drug must be metabolized physiologically in the human body, and other components of the FBB extract could alter its effects.

Additionally, the authors dismiss the fact that by pre-treating cells or mice in an in vivo model with FBB or Dex, they preprogram the cells. They then polarize the cells with LPS and again treat them with FBB or Dex, later arguing that FBB must be present earlier than LPS due to differences in efficacy. If they suggest that FBB can polarize cells toward an anti-inflammatory state, they must show direct evidence—such as demonstrating that FBB treatment upregulates anti-inflammatory gene expression in macrophages.

The authors also appear confused about the use of housekeeping genes, underestimating that, mechanistically, housekeeping gene expression should remain stable regardless of whether cells adopt pro- or anti-inflammatory phenotypes. The request was to show that FBB does not completely shut down pro-inflammatory signaling through targeting the PI3K, MAPK, and NF-κB pathways, but that other pathways may remain unaffected. This would demonstrate that FBB specifically targets these pathways while allowing cells to remain active in mitigating inflammation. In the interaction data from Figure 1C, several genes related to an anti-inflammatory phenotype—such as PPARγ, EGFR, RXR—are regulated. Note:IL8, a pro-inflammatory target, not an anti-inflammatory gene.

It has long been assumed that natural compounds have only beneficial effects, and numerous publications promote this perspective. However, the possibility of off-target effects or adverse reactions is often neglected.

In conclusion, all necessary control experiments must be included to properly evaluate the beneficial effects of FBB extracts.